# Diode laser based gas analyzer for the simultaneous measurement of $CO_2$ and HF in volcanic plumes

Antonio Chiarugi[1,2], Silvia Viciani[2], Francesco D'Amato[2], and Mike Burton[3]

[1]INGV, Pisa, Italy
[2]CNR - National Institute of Optics, Firenze, Italy
[3]University of Manchester, School of Earth and Environmental Science, Manchester, UK

*Correspondence to:* Francesco D'Amato (francesco.damato@ino.it)

**Abstract.** A portable analyzer, for simultaneous detection of $CO_2$ and HF emitted by volcanoes and fumaroles is described. The system is based on two fiber coupled distributed feedback lasers and only one multipass cell, and provides the absolute concentration values of the two gases, without requiring a calibration procedure, at a maximum rate of 4 Hz. The spectrometer can operate both in a closed-cell configuration and in an open-cell setup, with the latter mitigating problems associated with chemisorption of the HF molecule. The concept, practical realization and laboratory performance of the device are presented, together with results from a first test campaign measuring volcanic gases emitted by the crater of Vulcano, Italy. We obtained an in-field sensitivity of 320 ppb for $CO_2$ and 20 ppb for HF at 2 s integration time.

## 1 Introduction

The dynamics of magma storage and ascent are reflected in the composition and flux of gases released by active volcanoes (Symonds et al., 1994; Giggenbach, 1996; Allard et al., 2005; La Spina et al., 2010). The major volatile species, $H_2O$, $CO_2$, $SO_2$, HCl, HF all have pressure-dependent solubility profiles, producing significant variability in gas composition as a function of pressure in the magmatic system (Lesne et al., 2012) which is then further modulated by the degassing style and magma dynamics (La Spina et al., 2016). The viscosity of magma controls its flow dynamics, and this is in turn controlled by the dissolved water contents of the magma and vesicularity, producing complex non-linear relationships between magma ascent and degassing (Gonnermann et al., 2003).

The thermodynamics of phase changes from dissolved to exsolved volatiles and crystal formation, together with adiabatic expansion, all contribute to the evolution of magmatic temperature (Blundy et al., 2006; La Spina et al., 2015), providing a further feedback mechanism as viscosity is also highly dependent on temperature. Unravelling this complex behaviour is a fundamental aim of volcanological science, and a key underpinning for this research is precise and accurate measurement of gas compositions at the surface. A further important scientific outcome from such measurements is new insight into atmospheric chemistry processes occurring within volcanic plumes. The aim of this work is to present results from a novel new instrument designed to be usable from airborne platforms.

A variety of methods have been developed to measure volcanic gas compositions, all with individual strengths and weaknesses. HF is measureable with filter packs (Pennisi et al., 1998; Mather et al., 2004; Allard et al., 2016), diffusion tubes (Rüdiger et

al., 2017), and direct sampling by bubbling through alkali solutions (Giggenbach et al., 1991; Wittmer et al., 2014), all of which require post-collection analysis in a chemical laboratory. Open-path Fourier transform spectroscopy (Burton et al., 2000; Duffel et al., 2001, 2003; Horrocks et al., 2003; Burton et al., 2007) measures both $CO_2$ and HF directly in the field. In-situ sensors such as MultiGas (Aiuppa et al., 2005; Shinohara et al., 2005, 2008; Aiuppa et al., 2008; Roberts et al., 2017) have allowed automated measurements of plume gas compositions, primarily $CO_2$ and S species, but not HF to date, and very fast changes in gas composition are challenging to quantify due to the slow and differing frequency response of commonly used optical $CO_2$ and electrochemical S sensors. Drifting calibrations make traceability of long term installations challenging. Remote sensing using infrared (Mori et al., 1997; Francis et al., 1998) or ultraviolet spectroscopy (Galle et al., 2002; Edmonds et al., 2003; Galle et al., 2010; Butz et al., 2017) is effective for safe determination of plume gases, and has a key capacity to measure gas compositions during explosive activity, but requires large path-integrated concentrations of gas. This is particulalry critical for the rapid measurements of quickly changing gas concentrations which are envisaged in an airborne sampling context. Tunable diode laser-based in-situ spectrometers may overcome the challenges of chemical sensors, allowing traceable, accurate, precise and high frequency measurements, and many volcanic gases have accessible near- and mid-infrared absorption spectra. They can also work at high frequency ($> 1\,$Hz), which is a requirement for airborne surveys where rapid changes in gas concentration are common. However, TDL instruments are usually bulky, delicate and ill-suited to the rigours of volcanogical fieldwork. A final essential requirement for volcanic gas sensing is that multiple gases are measured in the same volume of gas at the same time, whilst avoiding chemisorption processes. This allows the direct determination of volcanic gas molar ratios, which are the key to unravelling volcanic processes. $SO_2$ is relatively easy to quantify due to a strong UV absorption spectrum and low background concentration, allowing straightforward $SO_2$ flux quantification (Stoiber et al., 1983; Galle et al., 2003; Mori et al., 2006). So, the knowledge of $SO_2$ flux, when added to the in-situ measurement of the ratios among $SO_2$ and other gases, extends the information about flux to these gases too.

In the context of the European Research Council project CO2Volc, which has the aim of improving our understanding of global emissions of $CO_2$ from volcanoes, we identified the need for a custom-built TDL platform for volcanological applications. Here, we report on the first measurements of $CO_2$ and HF we conducted with the first analyzer of the planned CO2Volc TDL platform, which was tested in the laboratory and on fumaroles on Vulcano island, Italy. The species $CO_2$ and HF are ideally suited for revealing magma dynamics, being two end-members for solubilty in magmas; $CO_2$ is highly insoluble, and begins to exsolve from magma at depths typically $> 10\,$km, while the bulk of HF degassing occurs at very low pressures.

## 2   State of the art for in-situ volcanic gas measurement

The requirements for a successful and effective volcanic gas analyser are stringent. The instrument must be compact and light for field portability, to allow easy field deployment with a variety of transport solutions, including backpacks, airplanes and drones. It must also be robust enough to allow operation in hostile and harsh environments, such as high humidity, fluctuating temperature and high concentrations of acid gases. Moreover, volcanic gas sensors would ideally have low power requirements to allow prolonged field measurements and the capability to work unattended through remote control or fully autonomously.

Finally these constraints must not limit instrumental performance, maintaining high selectivity to multiple molecules (in order to identify specific volcanic gas species), high sensitivity (in order to quantify small changes in gas concentration), high precision and accuracy (in order to measure low concentrations in dilute plumes) and response times of the order of 1 s (to permit airborne measurements and resolve rapid changes in gas composition). In addition to these requirements, the detection of acid

gases, such as HF, require further precautions due to the rapid chemisorption of acid molecules on surfaces of the instrument, precluding pumps and filters. Typically, the limits on what makes an instrument field-portable on a volcano are the carrying capacity of a group of two or three people. Thus, up to 10 kg for an instrument and a few kg for batteries is typically ideal. Several laser spectrometers have been developed to measure multi-species gas emissions (Richter et al., 2000, 2002) but these were poorly suited for the challenging volcanological field context, in particular for the key requirement of simultaneous mea-

surement of multiple gas species and for the problems connected with chemisorption. Several analyzers fulfill the requirements for $CO_2$ and $H_2O$ measurements (Gianfrani et al., 2000; Rocco et al., 2004). Several commercial instruments provide simultaneous detection of $CO_2$ and $H_2O$ (e.g. LICOR 7000 and LICOR 840A, URL = https://www.licor.com/env/products/gas_analysis/). This device yields an accuracy of 1% for both gases, with a RMS noise of 25 ppb for $CO_2$ and 2 ppm for $H_2O$ at 1 s integration time. Its weight is less than 9 kg, and the power consumption is within 40 W. However, this device requires that gases are

pumped through a narrow tube, which makes simultaneous measurement together with acid gases impossible.

Hydrogen fluoride could be measured using Cavity Ring-down Spectroscopy (CRDS) (Morville et al., 2004), but low pressure operation is needed which is extremely challenging to achieve without experiencing strong chemisorption. Several other commercial devices offer HF detection, but none of them fulfills all the requirements for volcanological applications. The simplest

ones are those used in human exposure monitoring (GfG Instrumentation, Model TN 2014 Micro IV, URL = http://goodforgas.com/shop/mic iv-single-gas-detector), but the response time is too long (∼90 s) and the cross sensitivity, in particular to hydrogen chloride (which is typically also present in volcanic plumes) is very strong. In commercial Fourier Transform Infrared (FTIR) spectrometers, as in Environnement S.A., Model MIR FT, URL = http://www.environnement-sa.com/products-page/en/emission-monitoring-en/multigas-stationary-monitoring-systems-en/mir-ft-2/?cat=120, the detection limit is rather high (about 180 ppb

with a time response of 2 s), and typically a multipass cell is needed for in-situ measurement, which can provide a chemisorption surface. A higher sensitivity (2 ppb) can be reached with Ion Mobility Spectrometry (IMS) (Molecular Analytics, Particle Measuring Systems, Model ProSentry-IMS HF Analyzer, URL = www.pmeasuring.com), but again weight (23 kg) and power consumption (more than 500 W) are too high. Very high sensitivity can be obtained with Off-Axis Integrated Cavity Output Spectroscopy (ABB - Los Gatos Research, Hydrogen Fluoride Analyzer, URL = www.lgrinc.com) (detection limit of 0.2 ppb

in 1 s, but with a power consumption of about 100 W and a weight of about 29 kg). Overall, we determined that none of these commercially available solutions were suitable for volcanological applications, as their weight, power and chemisorption characteristics don't fit our constraints. Furthemore, our specific requirements include measuring combinations of several volcanic gas species, such as $CO_2$ and HF, in the same volume of air, and devices like these do not exist on the commercial market. As we show here, the new instrument was designed to balance the competing requirements of low weight and power consump-

tion, physical robustness and the required selectivity, sensitivity and time response performance. We adopted direct absorption

in combination with a multi-pass cell as a detection technique which, though noisier and less sensitive than CRD or ICOS, allows a great simplification of the apparatus and doesn't need extremely high reflectivity of the cell mirrors. When measuring in the plume of a volcan, cleanliness of the mirrors is a serious concern.

In principle, detection limits and accuracy could be increased of one order of magnitude when adopting a detection technique based of laser frequency modulation (Corsi et al., 1999; D'Amato et al., 2002). Yet, this kind of detection raises severe concerns about calibration, in particular in an environment where temperature, pressure and mixture composition can vary significantly. This is because in these techniques the calibration relies on the stability of the lineshape which, on the contrary, is affected by the physical conditions of the measurement. So, also next developments of this class of instruments will adopt direct absorption.

The practical realization and the laboratory performance of the device will be described. Finally, we will show the results obtained during a first test campaign at the crater of Vulcano volcano (Aeolian Islands, Italy).

## 3  Experimental setup

The optical setup of the analyzer is based on two near-IR Distributed FeedBack (DFB) fiber coupled diode lasers and only one multipass cell (Fig. 1). In order to optimize the instrument size and to better exploit the available space, the laser sources and the entrance of the cell are on one side ("top") of the optical breadboard, while the reference arm, the detectors and the exit of the cell are on the other side ("bottom"), as in Fig. 2.

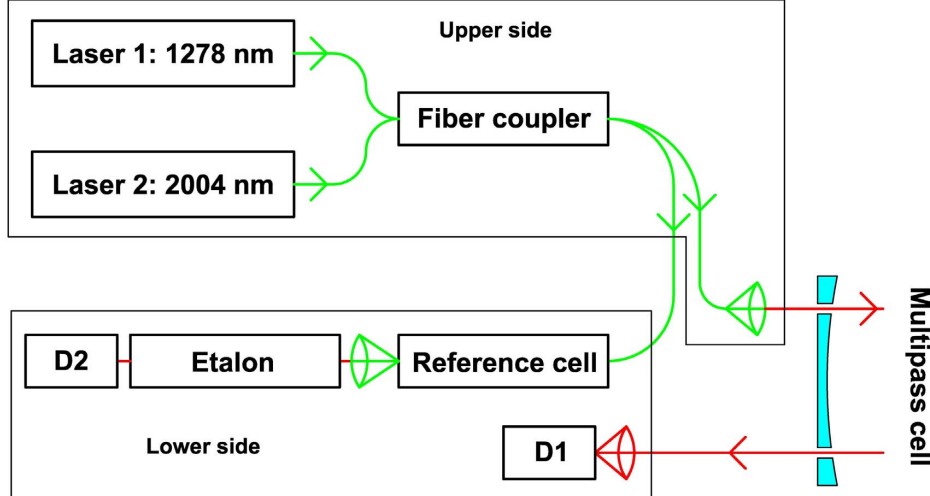

**Figure 1.** Optical scheme of the analyzer. Closed line encompass the components according to their locations, as per Fig. 2

On the top side of the breadboard, two fiber coupled DFB lasers emitting respectively at 1.278 $\mu$m and 2.004 $\mu$m (Eblana Photonics EP1392-DM and EP2000-DM, output power 5 mW and 2 mW) are directly mixed in a "2x2" fiber coupler, 50/50 transmission, working around 1.3 $\mu$m and 1.5 $\mu$m (Thorlabs 10202A-50-APC). As a matter of fact, this component is a stan-

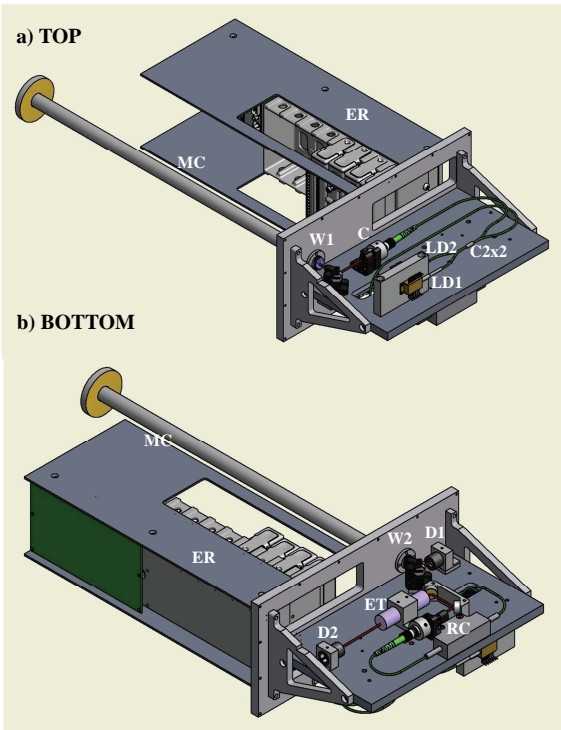

**Figure 2.** Setup of the analyzer. a) Top view. LD1 and LD2 diode lasers; C2x2: fiber coupler 2x2 50/50 transmission; C: collimator; W1: CaF$_2$ window for the entrance of the multipass cell MC. b) Bottom view. RC: reference cell; ET: BK7 etalon; W2: CaF$_2$ window for the output of the multipass cell MC; D1 and D2 detectors. ER: rack for electronics

dard one and in principle is not dichroic for the two laser wavelengths. Nevertheless we don't need an exact splitting ratio, and the attenuation of the coupler at 2 $\mu$m is low enough to allow us to use it. So about half of the power of both lasers is collimated and sent into the multi-pass cell, while the other half of the power of the two beams is sent towards the bottom side of the breadboard.

5   The multipass cell (Fig. 3) is a home made Herriott cell (Herriott et al., 1965) with a total pathlength of 20.23 m in 52 passes. It has two quartz mirrors mounted on a carbon fiber pipe. The pipe is kept by a stainless steel holder, which can be fixed to the analyzer body in a repeatable way by means of three spines. This allows a fast and easy mounting and removal of the cell in case of cleaning or replacement. As several identical cells were aligned in the same optical setup, in case of replacement no realignment of the analyzer is necessary, and just the orientation of the final mirror may require optimization. The beam

10   enters the multipass cell and gets back into the analyzer across two CaF$_2$ windows, tilted by an angle of 23° in order to avoid interference fringes.

On the bottom side of the breadboard, the beam exiting from the multipass cell is sent to a 20 mm plano convex lens and

focused onto a main detector (Hamamatsu G12183-010K, InGaAs PIN, 0.9-2.6 $\mu$m, $\phi$ 1 mm, uncooled), which is suitable for both laser wavelengths.

The other beam exiting from the 2x2 coupler is sent across a reference cell (manufactured by Wavelength References), containing both HF and $CO_2$, to verify the right settings of the two lasers, in particular for HF, which is usually absent in the atmosphere. The reference beam is then collimated and sent through a custom-built etalon in order to obtain a relative frequency reference for the linearization and calibration of the frequency scale. The etalon is made from BK7, 6 cm long and with Free Spectral Range (FSR) 0.0554/0.0558 cm$^{-1}$ (1662/1674 MHz) at 1278 and 2004 nm, respectively. Finally, the reference signal is measured with a detector identical to that used in the measurement channel.

The electronic part of the instrument is placed next to the cell (as shown in Fig. 2), and consists of a CompactRIO by National Instruments (cRIO 9074), which combines a dual-core processor, a reconfigurable FPGA (Field Programmable Gate Array), and five commercial plug-ins for fast acquisition (four independent channels, 1 MHz at 16 bits), slow acquisition (8 multiplexed channels, 500 kHz at 12 bits), digital I/O, thermocouple reading and data storage. Laser current drivers, laser temperature controllers and preamplifiers for detectors are home made.

In order to protect the optics and electronics against the volcanic gases, they are enclosed within a plastic cover, sealed with a rubber O-ring on the aluminum breadboard of the instrument. This means that HF is measured only in the volume of the multipass cell, but $CO_2$ is inside the plastic cover too. However, due to the choice of optics fibers components, inside the box the main beam travels in the air only from the collimator to the first CaF$_2$ window, and from the second window to the main detector. This pathlength is in total 11.5 cm, to be compared to about 20 m. So by neglecting this additional path, we overestimate the ambient concentration of $CO_2$ by $6 \cdot 10^{-3}$ or about 2 ppm. The relative effect is smaller when measuring $CO_2$ concentrations above ambient.

The device can operate in an open-cell configuration or in a closed-cell configuration. In the first case, we don't need any sampling air mechanism and the acquisition time is not limited by the time necessary to completely refresh the air inside the cell volume. In the latter, the cell is closed with a teflon tube and the air is sampled by a rotary pump, which provides an average flow of 20 l/min. So the 1 liter volume of the multipass cell is flushed in about $\sim 3$ s.

The pressure and the temperature inside and outside the cell are measured respectively by a silicon piezoresistive pressure sensor (Motorola, MPX2100A, with accuracy of about 1%) and by a PT100 sensor (National Instruments, NI 9217 RTD, with accuracy of about 0.8%).

The total power requirement for the normal operation of the spectrometer is about 20-25 W (in the open-cell configuration without pump) and 50 W (in the closed-cell configuration with pump). The power is provided by one (open-cell configuration) or two (closed-cell configuration) 4 cell LiPo battery, 6800 mAh at 14.8 V.

The instrument is robust and compact (with size of 60.5x27.6x13.2 cm), with a weight of about 8-9 Kg (pump and batteries included), which makes it particularly suitable as a portable instrument for in-situ operation in a hostile environment such as a volcanic area. The spectrometer can work for about 4-5 hours completely unattended and be remotely controlled via WiFi

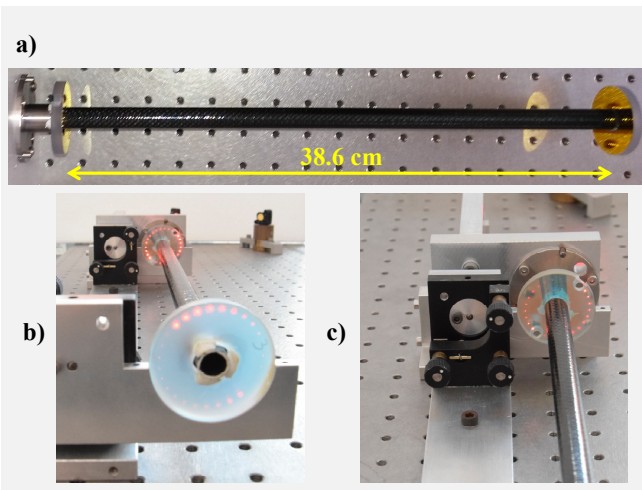

**Figure 3.** Home made multipass cell in the open-cell configuration (a). Photo b) shows the whole cell with the spots of the alignment laser onto the mirrors, photo c) is a detail of the entrance/exit mirror.

from outside the area of toxic gas emission.

## 4 Acquisition technique and data processing

In the acquisition procedure the two laser emission frequencies are scanned respectively across the selected absorption lines of the target molecules, by modulating the laser current with a ramp signal. The 2 $\mu$m-laser scans over the $CO_2$ absorption line at 4989.97 cm$^{-1}$ (2004.02 nm), while the 1.3 $\mu$m-laser is scanned over the HF absorption line at 7823.82 cm$^{-1}$ (1278.148 nm). The maximum tuning of the two lasers provides two frequency scans of 0.8 cm$^{-1}$ (24 GHz) around the $CO_2$ line and 1.5 cm$^{-1}$ (45 GHz) around the HF line, respectively.

The two signals measured by the main detector (D1) and by the reference detector (D2) are acquired synchronously on 2 acquisition channels of the CompactRIO at 1 MSample/s with a resolution of 16 bits. The two main and reference signals, sampled each with 4000 points per scan, are averaged 25 times and saved for a post-processing. Typical in-field main and reference signals are shown in Fig. 4.

The two laser sources work in sequence and are switched on alternatingly: when one laser is switched off, the current of the other laser is modulated, and conversely. As shown in Fig. 4, the modulation signal of each laser consists of 3 parts: (i) an initial interval, with a duration of 100 $\mu$s, during which the laser is turned off to get the background signal of the detector when no laser power is incident on it (in the following we refer to this signal as "zero-power signal"); (ii) an interval of 300 $\mu$s during which the laser current is maintained fixed at the starting value of the ramp, to allow the stabilization of the signal

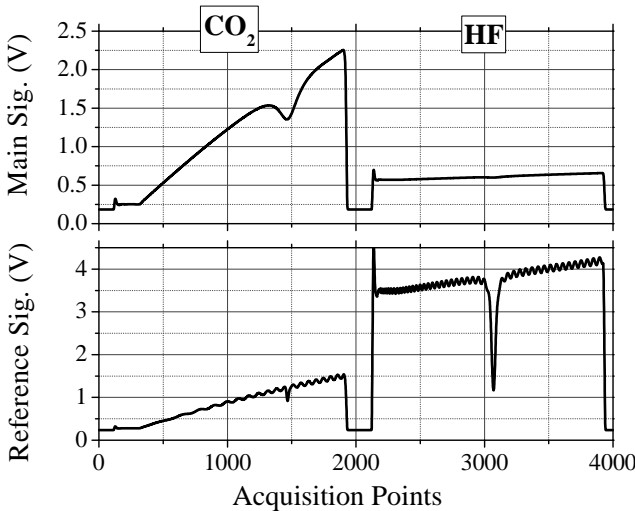

**Figure 4.** Typical signals recorded by the measurement detector D1 (top) and reference detector D2 (bottom) during a test campaign at Vulcano volcano.

before starting the frequency scan; (iii) the linear ramp with a duration of 1.6 ms. The repetition frequency for the complete sequence of modulations is about 250 Hz. If the time for averaging and saving data is also considered, the integration time for the absorption spectra of $CO_2$ and HF is 0.25 s. The time interval between the two laser scans is 2 ms, so that the two measurements can be considered simultaneous for the purposes of our applications.

The concentration values of the two gases are calculated by the absorption signal registered by the main detector (upper side of Fig. 4), which include the two absorption spectra of the target molecules, and by and measured pressure and temperature, according to the Beer-Lambert law. For each acquisition the zero-power signal can be subtracted, so that the absorbance can be derived independently from the absolute laser power. Consequently we don't need to know exactly the splitting ratio of the beam splitter or any kind of variability in the laser power. Moreover the changes of reflectivity of the mirrors in the multi-pass
cell, related to the interaction with the external ambient, do not influence the measured absorbance value except for affecting the signal to noise ratio.

The reference spectrum (bottom side of Fig. 4) consists of two kind of signal: the interference fringes due to the etalon and the absorption lines of the two molecules. Known the etalon FSR's, the interference signal allows to obtain a relative frequency reference for the calibration of the frequency scale. The presence of the absorption signals is necessary only to
check that the laser emission frequencies don't change over time. This is particularly important for HF which is usually absent in the atmosphere, since, without reference cell, it is impossible to verify the stability of the laser emission frequency and its overlapping with the selected absorption lines.

Here we present a quick description of the data processing, since a more detailed analysis has already been reported (Viciani et al., 2008). The absorption spectra, after subtracting the zero-power signal, are fitted with the exponential of a Voigt profile
multiplied by a second-order polynomial, which simulates the sloping laser power due to the ramping of the driving current.

When necessary, also one or more sinusoidal curves are included in the fitting function to take into account the presence of optical fringes. We use as Voigt profile the four-Lorentz Puerta-Martin approximation (Puerta et al., 1981; Martin et al., 1981), where the only free fitted parameters are the line amplitude and center frequency, while the Lorentzian and Gaussian half widths at half maximum (HWHM) are maintained fixed. The values of the HWHM are calculated as a function of temperature and pressure, measured for each acquisition, and of the molecular parameters (airbroadening coefficient and linewidth temperature coefficient) according to the HITRAN database (Rothman et al., 2013). Obviously, the frequency calibration of the x-axis becomes essential for this approach so that, for each saved absorption spectrum, the corresponding reference signal is recorded and fitted according to the etalon transmission equation, multiplied by a second order polynomial and for the absorption signal of the reference cell. According to the fit results, and knowing the etalon FSR's, the frequency scale can be determined for each acquisition.

The integrated absorbance for the two molecules is calculated according to the fitting parameters of the Voigt function, and the concentration $N$ is calculated from the absorbance knowing the multipass path length and the gas line-strength of $CO_2$ and HF, according to the HITRAN database (Rothman et al., 2013). Finally, the mixing ratio $MR$ is obtained according to the Equation:

$$MR = \frac{N}{N_0} \frac{T}{T_0} \frac{P_0}{P} \tag{1}$$

where $N$ is the calculated molecules concentration in $cm^{-3}$, P and T are the measured values of pressure (in atm) and temperature (in K), $P_0$=1 atm, $T_0$=296 K and $N_0$=2.470x$10^{19}$ $cm^{-3}$.

## 5 Laboratory performance of the instrument for the $CO_2$ channel

In order to test the laboratory performance of the instrument and to verify the reliability of the inferred concentration values, a mixture of known concentration was used in the closed-cell configuration. Such kind of test, if done with an HF mixture, would be deeply influenced by the property of the acid HF to stick, inside the tube closing the cell or inside the pipes connecting the multipass cell to the tank of the mixture. We also verified (as shown in the next section) that, by using teflon sampling pipes, the problem is reduced but not completely resolved, due to chemisorption of HF to water vapor film or dust covering the inside of the tubes. Consequently, we decided to carry out this laboratory test only for the $CO_2$ channel and to evaluate the performance of the HF channel directly in-field by using the open-cell configuration.

The measurement was carried out by introducing into the multipass cell a constant flow (about 0.3 l/min) of calibrated mixture (591$\pm$ 3 ppm $CO_2$ in synthetic air) at atmospheric pressure and also at 700 mBar, to simulate a typical pressure at the top of a 3000 m volcano (e.g. Etna). For the measurements below room pressure we used a scroll pump and two needle valves, at the entrance and exit of the multipass cell, to set the pressure at the desired value.

We show in Fig. 5 a normalized absorption spectrum of the $CO_2$ mixture, detected by our instrument at the maximum rate of 4 Hz, at atmospheric pressure and at a temperature of about 295 K, together with a fitted absorption line which includes optical fringes.

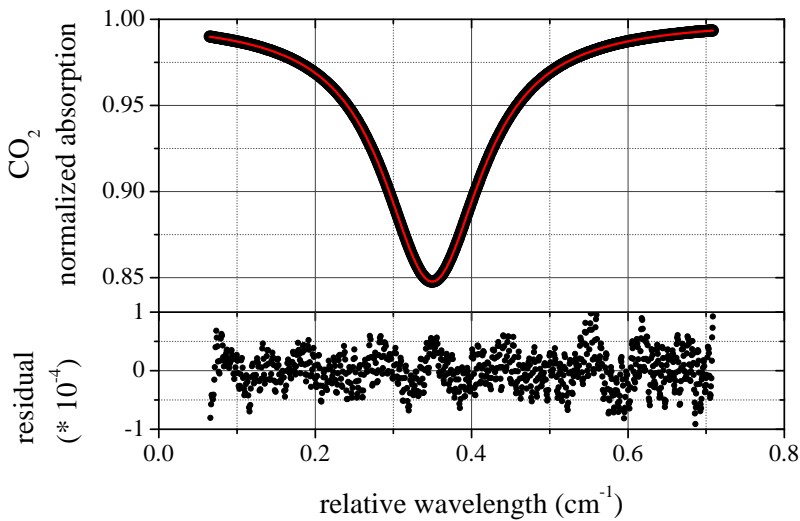

**Figure 5.** Normalized absorption spectra of a $CO_2$ calibrated mixture (of about 590 ppm in synthetic air) around 4989.97 cm$^{-1}$ (2004.02 nm) and Voigt fit results (red line). The residuals of the fitting procedure are shown in the bottom plot. The integration time is 0.25 s. The measurement was carried out at ambient pressure and at a temperature of 295 K.

In order to determine the accuracy of the measurement we followed the same formalism described in (Viciani et al., 2008).
The mixing ratio uncertainty is is determined by combining the accuracy of the temperature and pressure measurement (respectively 1% and 0.8%), the accuracy of the $CO_2$ line strength according to the HITRAN database (between 1% and 2%) (Rothman et al., 2013) and the uncertainty in the fitting procedure (0.2%). The resulting total accuracy is $< 4\%$.
From the spectra of Fig. 5 we retrieve a $CO_2$ mixing ratio of $(592 \pm 12)$ ppm. This value is in agreement with the concentration of the calibrated mixture.
In order to evaluate the long-term stability of the instrument, we repeated the same measurement, at a rate of 4 Hz, for about 1 hour, in both conditions of atmospheric pressure and about 700 mBar. Assuming as precision the standard deviation $(2\sigma)$ of the obtained mixing ratio values, we infer a $CO_2$ precision for 1-hour measurement of 0.1% (600 ppb) at atmospheric pressure and of 0.03% (200 ppb) at 700 mBar. The lower precision at ambient pressure is due to the fact that the absorption signal is as broad as the laser scan and it is not easy for the fitting procedure to clearly identify the parameters of both the second-order
polynomial which describe the background and the optical fringes. When the pressure is reduced and the absorption signals are narrower, the fitting protocol becomes more precise.

In the present case, the sensitivity of the instrument, defined as the minimum variation in the mixing ratio detectable by the instrument, is entirely determined by the precision. In order to evaluate the ultimate sensitivity of the $CO_2$ channel, an Allan-Werle Variance analysis of the obtained mixing ratios was carried out. An Allan-Werle Variance plot of the $CO_2$ measurements is reported in Fig. 6 as a function of the integration time.

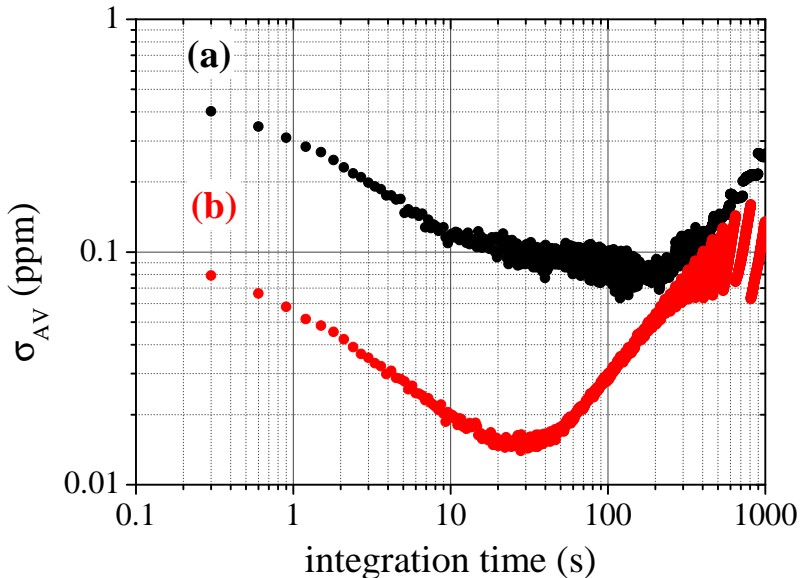

**Figure 6.** Allan-Werle Variance plot of 1 hour in-flow measurements of a $CO_2$ calibrated mixture of about 590 ppm in synthetic air, at a rate of 4 Hz in two different conditions: for a pressure of 985 mbar and a temperature of 295 K (curve a) and for a pressure of 687 mbar and a temperature 294 K (curve b).

We can conclude that the $CO_2$ sensitivity at the fastest integration time of the instrument (250 ms) is about 500 ppb at atmospheric pressure and less than 100 ppb at lower pressure. For an integration time of 1 s, a sensitivity of about 300 ppb at atmospheric pressure and of about 60 ppb around 700 mbar, is obtained. The best achievable sensitivity for the $CO_2$ channel can be reached for 110 s of integration time (about 80 ppb) at atmospheric pressure and for 30 s of integration time (about 15 ppb) at 700 mbar. Again, we believe that this reduction of the best integration time at low pressures is due to the better capability of the fitting protocol to clearly identify the background signals and the optical fringes.

## 6    In-field performances of the instrument for $CO_2$ and HF channels

Two test campaigns were carried out in order to evaluate the performance of the instrument and in particular of HF measurement, as this is the most sensitive to chemisorption processes.

The first campaign was performed in April 2015 at the crater of Vulcano volcano (Aeolian Islands, Italy). During this campaign the instrument was deployed in the closed-cell configuration, with ambient pumped through a 1 m Teflon tube into the cell (as in Fig. 3b). The integration time for each measurement was 1.5 s. We placed the instrument downwind of degassing fumaroles, producing exposure to a variety of gas concentrations.

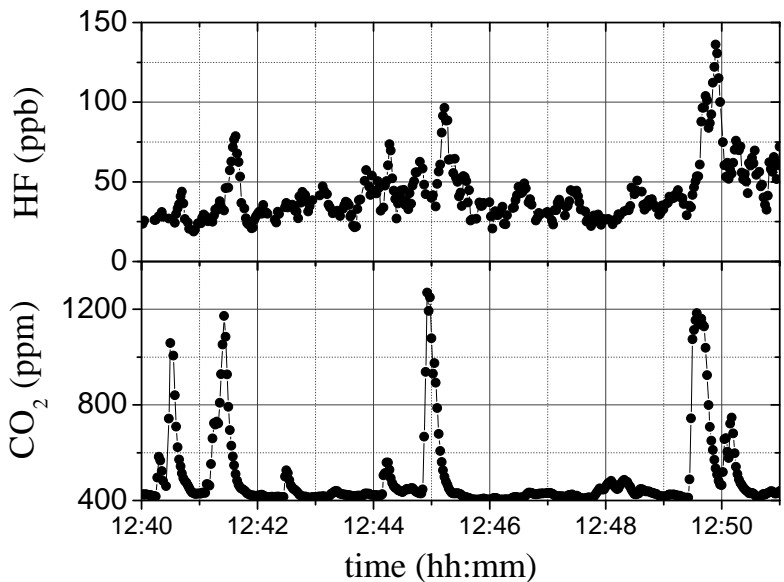

**Figure 7.** Time series of $CO_2$ and HF mixing ratios measured at the crater of Vulcano volcano on the 22nd April 2015, performed by the spectrometer in the closed-cell configuration. The measurements were carried out in an average condition of 990 mbar of pressure and 300 K of temperature. The integration time is 1.5 s. The delay in the integration times of the HF peaks with respect to the $CO_2$ peaks is due to chemisorption on instrument components in the closed cell configuration.

5    The results obtained for about 10 minutes of concentration measurement of HF and $CO_2$ are reported in Fig. 7. A correlation between the concentration peaks of different gases is observed, but the peaks are not well aligned in general, and the HF peaks show a significant delay compared with $CO_2$. Correlation analysis of the two concentrations clearly shows that a maximum correlation between the two gases is reached for a delay of about 16 s. Moreover, the correlation is poor with a maximum correlation coefficient of only 30%. We attribute these observations to chemisorption of HF molecules, which stick inside the
10  pipes or inside the tube covering the cell, in spite of the use of teflon components. When a gas cloud arrives to the pipe inlet, part of the HF is lost as it reacts with the walls, producing a delayed HF peak. The only way to completely solve this problem is to remove all the pipes and the tube covering the cell.

The open-cell configuration was employed in a second campaign, also at the crater of Vulcano volcano, performed during May 2015.

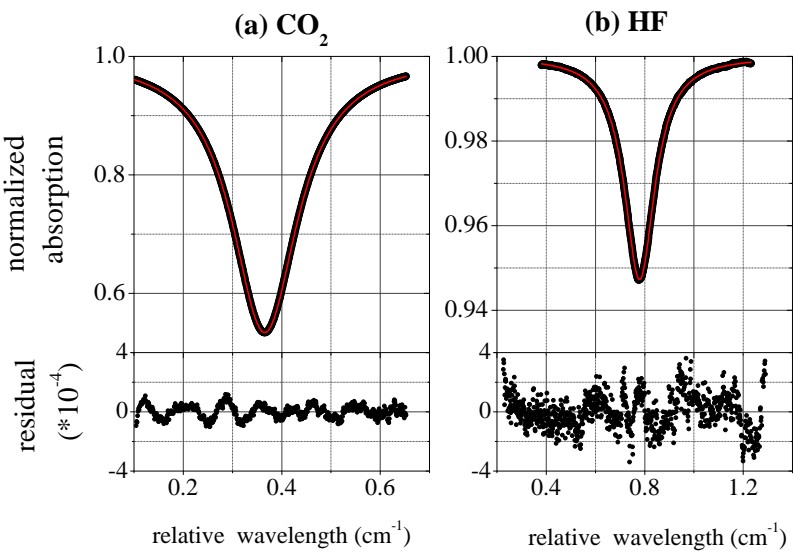

**Figure 8.** Typical normalized absorption spectra of $CO_2$ (a) and HF (b) detected in proximity of a fumarole of the Vulcano volcano on the 20th May 2015 and Voigt fit results (red line). The residuals of the fitting procedure are shown in the bottom plots. The $CO_2$ mixing ratio is 2350 ppm and the HF mixing ratio is 4.5 ppm. The integration time is 2 s. The measurement was carried out at a pressure of 990 mbar and at a temperature of 312 K.

A typical normalized absorption spectrum for both $CO_2$ and HF, with 2 s integration time for each measurement, is shown in Fig. 8. The measurement was carried out close to a fumarole of the volcano, with a pressure of 990 mbar and a temperature changing between 310 K and 320 K. The results of the Voigt fitting procedure and the corresponding residuals are also shown in Fig. 8. Three optical fringes for $CO_2$ and two optical fringes for HF are also included in the fitting curve.

The accuracy of the mixing ratio values is determined by the same parameters described in the previous section: the uncertainties on temperature, pressure and $CO_2$ line strength are the same as in the laboratory test; the accuracy of the HF line strength according to the HITRAN database is between the 1% and 2% (Rothman et al., 2013); the uncertainty of the fitting procedure is 0.2% for $CO_2$, exactly the same as in the laboratory test reported in Fig. 5, and 0.7% for HF. The resulting total accuracy is <4% for $CO_2$ and <4.5% for HF. By assuming the worst uncertainty, the $CO_2$ mixing ratio for the spectrum of Fig. 8a is

$(2350 \pm 50)$ ppm and the HF mixing ratio for the spectrum of Fig. 8b is $(4.5 \pm 0.1)$ ppm.

  The advantage of the open-cell configuration, with respect to the closed-cell employed in the previous campaign, becomes obvious when a correlation analysis between the two gases is performed. The delay of the HF peaks with respect to the $CO_2$, observed in Fig. 7, is now absent and a maximum correlation between the two gases is obtained without introducing any delay. The $CO_2$ mixing ratio as a function of the HF mixing ratio during 13 minutes of measurement is reported in Fig. 9 and

the linearity indicates a high correlation between the two gases, corresponding to a correlation coefficient of 95%. This high

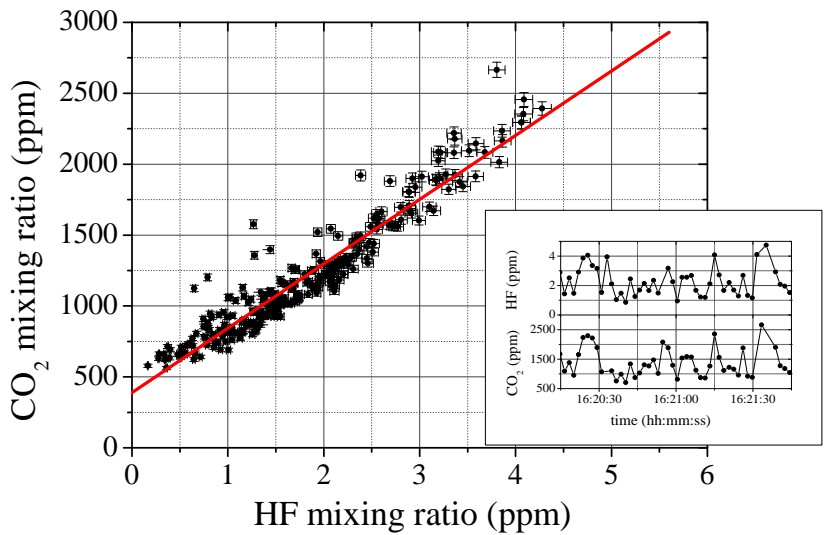

**Figure 9.** $CO_2$ mixing ratio as a function of the HF mixing ratio for 13 minutes of measurement in proximity of a fumarole of the Vulcano volcano on the 22nd April 2015. The red line is the result of a linear fit. The error bars are the accuracy of the mixing ratio values. The measurement was carried out in the open-cell configuration at a pressure of 990 mbar and at a temperature variable between 310 K and 320 K. The inset shows a zoom of the time series of $CO_2$ and HF mixing ratios during a 3-minute interval.

correspondence is better shown in the inset of Fig. 9, where a zoom of the simultaneous time series of $CO_2$ and HF mixing ratios during a 3-minutes interval is displayed.

A linear fit of the data of Fig. 9 allows to infer both the $CO_2$ ambient mixing ratio not due to volcanic emissions (calculated as the $CO_2$ value when the measured HF is zero), and the ratio between the $CO_2$ and HF mixing ratios. The obtained results are

$(390 \pm 20)$ ppm for the $CO_2$ ambient mixing ratio and $(460 \pm 50)$ for $CO_2$/HF ratio. These data fit well with the global trend for $CO_2$ concentration, see for instance the NOAA website (URL = https://www.esrl.noaa.gov/gmd/ccgg/trends/global.html). Our measurements of $CO_2$/HF with a molar ratio of $570 \pm 30$ were performed downwind of the F0 fumarole on Vulcano, so we expect most of the measured gases to be sourced from here, however we cannot exclude mixing with other fumarolic sources. This allows a comparison with measurements collected with OP-FTIR on F0 fumarole in 2002 (Aiuppa et al., 2004), which

revealed a $CO_2$/HF molar ratio of $175 \pm 20$. This difference may reflect either a change in gas composition from fumarole F0 or a potential contribution from different fumarolic sources in each measurement. In order to evaluate the in-field sensitivity of the instrument we assume as signal-to-noise ratio S/N the ratio between the normalized absorption signal and two times the standard deviation $(2\sigma)$ of the residual corresponding to the Voigt fit.

In Fig. 10 the S/N as a function of $CO_2$ and HF mixing ratios is reported. We use as noise value for the data of Fig. 10 the

mean value of the noise related to different fit results of the detected spectra. The mean noise value is $8.5 \ 10^{-5}$ for $CO_2$ and

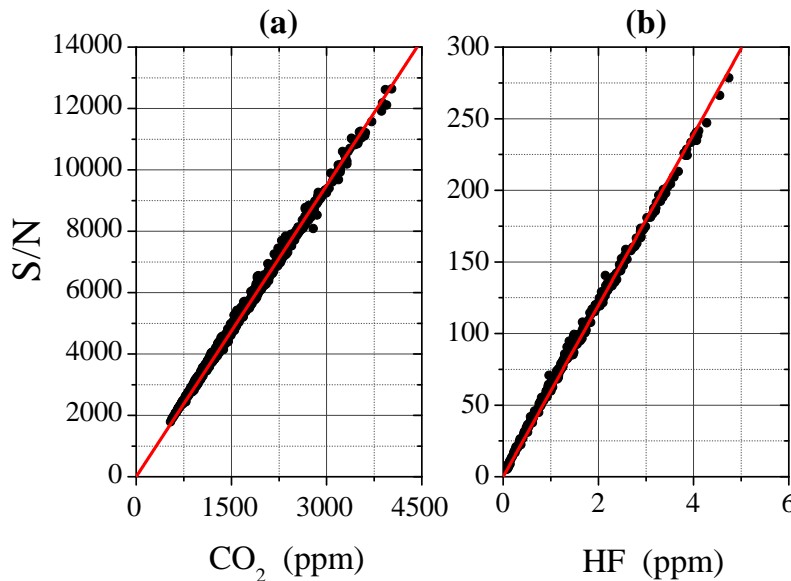

**Figure 10.** Signal-to-noise ratio S/N as a function of $CO_2$ mixing ratio (a) and HF mixing ratio (b) for 13 minutes of measurement in proximity of a fumarole of the Vulcano volcano on the 22nd April 2015. The red line is the result of a linear fit. The measurement was carried out in the open-cell configuration at a pressure of 990 mbar and at a temperature variable between 310 K and 320 K.

2.4 $10^{-4}$ for HF.

The sensitivity for the two channels of the spectrometer can be obtained as the variation in the mixing ratio corresponding to a variation of the signal equal to the noise, and it can be inferred from the slope of the linear fit of the data in Fig. 10. The obtained sensitivity for the $CO_2$ channel, with an integration time of 2 s, is 320 ppb, which is only slightly higher with respect

to the value of 250 ppb obtained during the Allan-Werle Variance laboratory test shown in Fig. 6. We can conclude that the sensitivity performances of the $CO_2$ analyzer are not seriously degraded by the in-field operation and they are reduced only by a factor of 1.3. Consequently, from the laboratory Allan-Werle Variance analysis we can estimate an in-field sensitivity, at the fastest integration time of 250 ms, of about 650 ppb and an in-field ultimate sensitivity (at 110 s of integration time) of about 100 ppb, which can be further increased if we work at lower pressure (about 20 ppb for an integration time of 30 s at a pressure

of 700 mbar).

For the HF channel, a sensitivity of 20 ppb, with an integration time of 2 s, can be obtained from the linear fit of Fig. 10b. This performance can degrade over time as the mirrors of the multipass cell become dirty due to water, dust and gas emissions from the volcanic plumes, and the detected power is consequently drastically reduced. When the instrument operates very close to the fumaroles, in about 30 minutes the sensitivity of the spectrometer (for an integration time of 2 s) degrades to 2 ppm for

$CO_2$ and to 200 ppb for HF. On the contrary, if the instrument is operated in lower acid gas and aerosol concentrations, as for

instance on board an aircraft, it could run for long periods of time without significant degradation of the signal to noise ratio. In-situ degradation of the mirrors could be partially solved by keeping the mirrors clean with an air blade, where a light flux of air is blown through the carbon fiber pipe holding the mirrors. The air can be taken a few meters apart, far from the high concentrations at the emission points, and blown on the mirrors by using a pump with a dust filter at its inlet. Consequently, a small volume of air close to the mirrors will be filled with this purge air, where HF concentration is negligible and $CO_2$ concentration is close to its ambient level. This means that the effective distance between the mirrors is reduced. This reduction (estimated as lower than 0.3%) must be taken into account when concentration values are calculated. Moreover, for $CO_2$, the values of concentration before or after each emission peak could be used to correct the peak values.

## 7    Conclusions

We produced a new analyzer for the simultaneous measurement of the mixing ratios of $CO_2$ and HF in volcanic gas emissions. This device features low weight and power, as well as resistance to the harsh environmental conditions. The in-field spectrometer sensitivity, obtained during a campaign at the crater of Vulcano, is 320 ppb for $CO_2$ and 20 ppb for HF, for an integration time of 2 s. According to laboratory tests, this sensitivity decreases by about a factor 2 when the instrument is employed at its maximum rate of 4 Hz. However, the device performances improve when the measurements are carried out at a pressure lower than the atmospheric one. In particular the $CO_2$ sensitivity increases of about a factor 5 when the pressure is reduced to 700 mbar, a typical pressure at the top of a 3000 m volcano (e.g. Etna).

We are planning to extend the measurement to $H_2O$, by detecting water absorption close to a $CO_2$ one. This requires a higher tunability DFB laser at 2 $\mu$m, which is under procurement. Moreover we are designing a portable experimental platform for in-situ simultaneous measurements of 5 volcanic gases (HCl, $CO_2$, HF, $H_2O$ and $SO_2$ ). The platform will include two mid-IR spectrometers, one of which will be the instrument described in this paper, and an UV spectrometer. The platform will be employed for measurement campaigns in-field and on board aircraft.

In order allow deployment on board of drones, we will reduce size and weight by replacing the cRIO crate with a smaller electronics, namely Red Pitaya by StemLab, we will use only fiber-coupled lasers in the near infrared, so eliminating most mirrors and their holders, and we will use aluminum and carbon fibers only for those parts which strictly require mechanical hardness. All other parts will be realized by 3D plastic printing. As drones can fly much closer to plumes than manned aircraft, the concentrations to be measured are expected to be at least one order of magnitude higher than in the present measurements, and so the multipass cells will be shorter and lighter too.

*Acknowledgements.*  The activity was supported by the FP7-IDEAS-ERC Project CO2Volc (Grant 279802). The authors thank Mr. Marco De Pas, Mr. Mauro Giuntini and Mr. Alessio Montori for the realization of the non-commercial electronics, and Mr. Roberto Calzolai and Mr. Massimo D'Uva for the realization of mechanics. We greatly appreciate the support of Manuel Queisser, Giuseppe Salerno and Alessandro la Spina in performing the field measurements.

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
