# Peer review of "Diode laser based gas analyzer for the simultaneous measurement of $CO_2$ and HF in volcanic plumes"

_Atmospheric Measurement Techniques, 2017_

## Referee Comment (RC1) · C. Kern (Referee) · 7 Sep 2017

Summary

This manuscript describes a new instrument for measuring CO2 and HF abundances in volcanic gas. The instrument measures the absorption of light emitted by two tunable diode lasers (TDL) as it passes through an optical cell along a fixed path. The light path is extended beyond the physical length of the cell by multiple reflections on mirrors on either end of a multi-pass cell. A specific gas is measured by tuning the laser wavelengths across a characteristic near infrared absorption line of that gas and measuring the attenuation of the laser light after having passed through the measurement cell.

[Figure]

Though the operating principle is similar to that of some existing instruments, the authors have succeeded in adapting the TDL absorption technique to a confined sample space. The multi-pass cell allows in-situ measurements to be made with relatively high sensitivity and high temporal resolution. Such an instrument is ideally suited for airborne measurements, where an aircraft flies the instrument through the plume downwind of a volcano. In this study, the authors present the results of first test measurements performed on the ground in proximity to degassing fumaroles on Vulcano Island, Italy. These show that the instrument appears to be functioning as expected and in particular allow derivation of the detection limits, sensitivity, accuracy and precision of the instrument.

The manuscript is well-organized and easy to follow. Aside from a number of minor corrections, it would benefit from a more thorough literature review as well as a more detailed description of the instrument and the test measurement study site. These points are elaborated on in more detail below. After these issues are considered, I recommend this manuscript be accepted for publication in Atmospheric Measurement Techniques, as the work presented here represents a significant step forward in our ability to accurately measure certain major components of volcanic gas.

Specific issues

The manuscript would benefit from a more detailed literature review and inclusion of more relevant references to other, previously published work. In the introduction, it would be helpful to mention all the current methodologies that are typically used to measure HF in volcanic plumes: filter packs, diffusive tubes and chemical traps are missing, along with at least one reference for each. Direct sampling is also a means of measuring HF and CO2, and is mentioned, but no references are given. Clearly, none of these techniques perform the analysis in the field, so they are in some ways inferior to the new instrument described here, but they are the current standard means of performing these measurements and worth describing and referencing.

As far as I am aware, there is at least one commercial TDL-based instrument for measuring CO2 and HF. The BOREAL Gasfinder (http://www.boreal-laser.com/products/) can be ordered in CO2 and HF versions, and has both an open-path and in-situ measurement modes. I believe that the sensitivity of the in-situ measurements is likely inferior to that of the new instrument described here, as the measurement cell does not appear to have a multi-pass configuration. Also, the two species cannot be measured by the same instrument – two separate instruments are needed. However, due to the similar or even identical measurement principle of this commercial instrument to the prototype described here, it should probably be mentioned in the manuscript and the differences explained.

In general, the introduction might be improved by mentioning from the very beginning that the design goal of this instrument was to implement an instrument for use in airborne measurements. This would make it clear why many of the other techniques that are available already would not work well.

Since one of the main aspects of the manuscript is the description of the new instrument, the addition of a few more details would be valuable. For one, the 'optical scheme' shown in Figure 1 describes they physical setup of the individual components but does not describe the operation principle of the instrument very well. A schematic of the light path could help here. In this schematic, the path of light from each of the two lasers to the detectors could be followed. This would also augment the description in the text better than the current Figure 1. In the end, it is less important if a component is mounted on the top or bottom of the breadboard than it is to know which optical elements are passed in which order.

As it stands, the functionality of the reference channel did not become entirely clear to me. I understand that the etalon adds interference fringes to the reference signal which can then be used to calibrate the frequency scale of the measurement. But the light also passes a gas cell containing HF and CO2? Is the purpose of this just to ensure that the laser wavelength does indeed overlap with the absorption line of the

respective gases? Can't this also be ascertained from the etalon fringes themselves? Perhaps a recording from the reference channel can also be plotted, and the evaluation of this channel explained in more detail? It would be interesting to see what the etalon interference fringes look like. It is my understanding that the reference channel is only used for spectral calibration and not otherwise used in the retrieval of the CO2 or HF abundance. Is this correct?

It's interesting to see how the instrument's performance appears to improve slightly with decreasing pressure in the sample cell. Perhaps a bit more detail could be provided on how these underpressures were obtained? Was the pump pulling through a pinhole? Or was a more sophisticated flow controlled used?

Finally, it would be useful to include more details on the field experiments that were performed with the instrument at Vulcano Island. Where exactly was the instrument located (perhaps include a map?). More importantly, which fumaroles were being sampled? Vulcano is a well-studied field site. How do the results of the measurements compare with those obtained by others? See e.g. Aiuppa et al 2004, Intercomparison of volcanic gas monitoring methodologies performed on Vulcano Island, Italy and Inguaggiato et al 2012, Total CO2 output from Vulcano island (Aeolian Islands, Italy), but there are many other reports too. While it is not the main thrust of this manuscript to provide new data for volcanology, comparing the obtained results with others would strengthen the case that the instrument is performing as expected.

Minor corrections

Abstract, L4 – Consider changing 'remove all problems to 'mitigate problems associated with chemisorption'

P1, L20 – Please give references for examples of direct sampling and sampling via alkali solutions.

P1, L22 – Consider adding Aiuppa et al 2005, 'Chemical map- ping of a fumarolic field:

La Fossa crater, Vulcano island (Aeolian islands, Italy)' and Shinohara (2005), A new technique to estimate volcanic gas composition: plume measurements with a portable multi-sensor system to the references for MultiGAS.

P1, L23, consider omitting 'to be performed'

P1, L24, 'quantify due to THE slow and differing...'

P2, L1, consider replacing 'chemical-based' with 'electrochemical'.

P2, L2, Please add references for remote sensing via ultraviolet spectroscopy, e.g Galle, B., C. Oppenheimer, A. Geyer, A. J. S. Mcgonigle, M. Edmonds, and L. Horrocks (2002), A miniaturised ultraviolet spectrometer for remote sensing of SO2 fluxes: a new tool for volcano surveillance, J. Volcanol. Geotherm. Res., 119, 241–254. and

Edmonds, M., R. A. Herd, B. Galle, and C. M. Oppenheimer (2003), Automated, high time-resolution measurements of SO2 flux at Soufriere Hills Volcano, Montserrat, Bull. Volcanol., 65(8), 578–586, doi:10.1007/s00445-003-0286-x. and

Galle, B., M. Johansson, C. Rivera, Y. Zhang, M. Kihlman, C. Kern, T. Lehmann, U. Platt, S. Arellano, and S. Hidalgo (2010), Network for Observation of Volcanic and Atmospheric Change (NOVAC)—A global network for volcanic gas monitoring: Network layout and instrument description, J. Geophys. Res., 115, D05304, doi:10.1029/2009JD011823.

Also consider adding more references pertaining to IR spectroscopy.

P2, L3, Please define 'path amounts'. I assume you mean path-integrated concentrations.

P2, L4, I would argue that the technique described in this paper falls under the category of infrared spectroscopy and therefore is not 'poorly suited' for in-situ measurements.

P2, L4, I'm not sure what an 'in-situ spectrometer' is.

[Figure]

P2, L8, Ultraviolet spectroscopy mostly only measures sulfur dioxide. Therefore, it's not clear why sensitivity to multiple gases is an 'essential requirement for volcanic gas sensing'. Perhaps it's essential to in-situ measurements?

P2, L9, Consider omitting 'whilst avoiding chemisorption processes' here and discuss it later.

P2, L20, 'CO2 is HIGHLY insoluble. . .'

P2, L20, Please clarify what becomes saturated with what at depths larger than 10 km.

P2, L28, I believe that quantifying small changes in gas concentration requires high 'precision', not high 'sensitivity', correct?

P2, L29, Consider rewording to '(in order to RESOLVE RAPID CHANGES IN GAS COMPOSITION).

P3, L3 – The LICOR LI-840A also measures CO2 and H2O. Consider rewording to 'Several commercial instruments provide simultaneous detection of CO2 and H2O (e.g. LICOR 7000 and LICOR 840A).

P3, L13 (Environment) is not a valid reference. Please describe FTIR instruments in the text rather than in the reference list.

P3, L21, . . . on the COMMERCIAL market.

P3, L22, The design requirements for volcanological applications in general vary quite a bit depending on access to the volcano, instrument deployment platform, volcanic hazards, monitoring vs basic research, and other factors. Perhaps it's best to focus on the design requirements for airborne measurements of volcanic plumes here?

P3, L26, . . . less sensitive THAN CRD or ICOS,. . .

P3, L27, Consider rewording to 'source of concern DUE TO THE DESIRE FOR AN OPEN PATH CONFIGURATION'.

P3, L29, '. . . multipass cell, optimizes the instrument size and reduces the weight'

P3, L32, 'two GAS CONCENTRATIONS and THEIR RELATIVE ABUNDANCE'

P3, L34, '. . .values of the two gases can be provided at a maximum rate of 4 Hz without the need for calibration.'

P4, L3, '. . . laboratory PERFORMANCE of the device'.

P4, L3, Consider omitting 'with an Allan-Werle Variance analysis' here and discuss this later.

P4, L5, Consider omitting the detection limits here, as these are results and should be reported on later.

P5, L5, Consider replacing 'exploit' with 'use'

P5, L12, '. . . orientation of the final mirror MAY require OPTIMIZATION.'

P5, L15, '. . .multipass cell ARE sent. . .'

P5, L24, 'All electronics ARE placed. . .'

P5, L25, Please explain the acronym FPGA

P5, L30, 'In order to protect THE optics and electronics from volcanic gases. . .'

P5, L34, 'So by neglecting this ADDITIONAL path, we overestimate the ambient concentration of CO2 by 0.6% or about 2 ppm. The relative effect is smaller when measuring CO2 concentrations above ambient.'

P5, L34, Do you overestimate the path by 0.6%? If you know about this issue, why not simply correct for it?

P6, L6, 'the 1 liter volume'

P6, L12, '4 cell LiPo batteries'

P6, L13, '. . . a weight of about 8 kg (pump and batteries included), WHICH makes it particularly suitable as a portable instrument for in-situ operation in a hotile environment such as in a volcanic area.'

P6, L15, 'unattended and be remotely controlled via WiFi from outside the area of toxic gas emission.'

P6, L16, How far does the WiFi actually reach?

P6, L20, '. . . across a characteristic absorption LINE of the target molecules.

P6, L21, '. . .HF absorption LINE. . .'

P7, L2, replace 'alternatively' with 'alternatingly'

P7, L5-6, Replace 'a region' with 'an interval'

P7, L8, '. . . around the CO2 line and 1.5 cm-1 around the HF line, respectively.'

P7, L10, Please clarify what is meant by the 'zero-power signal'. I assume you mean the intensity measured on either side of the absorption line?

P7, L10, 'allows derivation of the absorbance independent of the absolute laser power. Consequently, the splitting ratio of the beam splitter and the reflectivity of the mirrors in the multi-pass cell do not influence the measured absorbance value except for affecting the signal to noise ratio.'

P7, L114, What is a '4000-points main signal'? Are you sampling the spectrum with 4,000 points per wavelength scan?

P7, L16, I understand that the measurement of CO2 and HF is nearly coincident for practical purposes of the measurement, but I would omit 'simultaneous' here because you just explained that the two gases are measured in alternating manner.

P7, L16, Omit 'of' before 0.25 s.

P7, L18, Again, please clarify what 'zero-power signal' means

P7, L18, '. . . ARE fit with . . . multiplied by a second order polynomial.'.

P7, L18, Are you sure you need to multiply by a second order polynomial? I would have expected the addition of a polynomial to described the ramping intensity. As mentioned before, it would be nice to actually plot a raw spectrum that includes the interference fringes (in the reference channel) and the ramping up of the laser intensity.

P7, L22, which 'molecular parameters' from the HITRAN database are relevant? Can you please be more specific?

P7, L25, 'multiplied BY a SECOND order. . .'

P7, L20-27, The HITRAN database contains line strength information, not absorption cross-sections. The line strength is defined as the integral of absorbance of a given line. Since this TDL instrument resolves the individual absorption lines of CO2 and HF, respectively, couldn't the line strength simply be measured as the integrated absorption over the measured wavelength interval? The advantage of deriving the line strength directly is that it should be independent of temperature and pressure, correct? Could you please explain why the four-Lorentz Puerta-Martin approximation is used rather than simply determining the line strength and calculating the column density and gas concentration from that?

P8, L5, '. . . laboratory performance. . .'

P8, L5-10, How do you know that chemisorption prohibits a laboratory test using a pre-mixed HF calibration gas? Did you attempt the experiment? Can you provide a reference from the literature as to why such an experiment would surely fail? I would think that dust and water vapor in the sampling apparatus could be avoided in a laboratory setting.

P8, L11, '. . .performance of the HF channel. . .'

P8, L12 and throughout the manuscript: Consider running a global 'search and replace' on the entire manuscript and replace all instances of 'has been' and 'have been' with

'was' and 'were' unless referring to studies that were conducted previous to the work presented here.

P8, L14, How was the under-pressure of 700 mBar achieved?

P8, L17, I don't understand why two optical fringes need to be included in the fit procedure when fitting the measurement channel. The reference channel passes the etalon, so it makes sense that fringes would appear there, but why are they included in the fit for the measurement channel? Also, they are not obvious in figure 3. Is that because they are very low-amplitude compared to the absorption line?

P8, L28, Please replace 'lower pressure' with the actual pressure that was used (700 mBar?)

P8, L30, Please explain why a narrower line shape improves the identification of the background intensity, and why optical fringes are present in the measurement spectrum.

Figure 3 caption, '. . . in the bottom plot.'

Figure 3 caption, and throughout the manuscript: When taking the average of multiple acquisitions to improve the signal to noise ratio, the term 'integration time' is typically used to describe the total time for all the individual acquisitions. The 'acquisition time' usually describes the time for just one acquisition. Consider using this verbiage throughout the manuscript.

P9, L1, 500 ppb appears to be the precision of the instrument, not the sensitivity of the instrument.

P9, L2, Detection limit and precision are not quite the same thing. In this case, since you will always have at least 400 ppm $CO_2$ in the ambient air, the detection limit is not really relevant I guess. Instead, the precision is what will determine if a small volcanic plume can be detected or not.

P9, L6, please explain what is meant by 'get rid of fringes'.

P9, L12, Please include more details on where the measurements were made, perhaps even a map?

P10, L6, Consider replacing 'emission peak' with 'gas cloud'

P10, L7, '. . . the only way to completely solve this problem. . .'

Figure 5 caption, '. . . is due to chemisorption on instrument components in the closed-cell configuration.'

Figure 6 caption, 'bottom plots'

Figure 6 caption and throughout the manuscript: Please be a little bit more precise when using the term 'concentration'. Concentration is a measure of the number of trace gas molecules in the sample cell or in a known volume of air. A typical unit is molecules / cm3. A mixing ratio is the ratio of trace gas molecules to total air molecules in a given volume. Typical units are ppm or ppb. The TDL instrument measures absorbance, which can be converted into column density (units of molecules / cm2) using the Beer Lambert Bouguer Law. This can further be converted into an average concentration in the cell. However, converting to a mixing ratio requires knowledge of temperature and pressure, as described in your equation 1. Since you are measuring temperature and pressure and correcting for these effects, it's fine to report mixing ratios, but please don't call them 'concentrations.'

P12, L6-8, Please clarify how you obtain the mean value of the noise from the residual. It appears that the residual contains some systematic structures as well as some statistical noise. How do you separate the two? Do you have any ideas where the systematic structures come from?

P12, L10, Again, do you mean precision rather than detection limit?

P12, L13, '. . . a factor of 1.3'
Figure 7 caption, Again, this figure plots mixing ratios, not concentrations.

Figure 7 caption, '. . . a 3-minute interval.'

P13, L6, Perhaps rephrase this sentence to 'If the instrument is operated in lower acid gas and aerosol concentrations it could likely run for long periods of time without significant degradation of the signal to noise.'

P13, L14, Do you mean 3 per mil or 3 percent? Since percent are used throughout the manuscript, consider replacing this with '0.3%' if that is the appropriate value.

P13, L15, 'emission peak could be used to correct'

Figure 8, again plots mixing ratios rather than concentrations

Figure 8, If I understand correctly, then the estimated detection limit of the instrument for CO2 was determined by extrapolating the measurements shown in Figure 8 a down to a S/N of 1. If this is true, this estimate would appear to have quite a large uncertainty given that no measurements were performed with S/N less than about 1800. If you want to peruse this methodology, please give the uncertainty of the detection limit obtained in this manner. In my opinion this is of somewhat limited use, however, as no measurements will ever be performed at less than about 400 ppm CO2 which corresponds to a S/N of more than 1,000. For the CO2 channel, the precision of the instrument is much more important than its detection limit.

P14, The manuscript lacks a 'Conclusions' section. The first paragraph on page 14 seems to belong to the conclusions so this might be an appropriate place for this heading.

P14, L8, '. . . by detecting water absorption. . .'

P14, The manuscript only includes a single sentence about future work. Given that this document only describes the very first tests of a new instrument, I would expect much more future work planned for the system. For example, I'm sure the system

is to be (or actually already was) run on an aircraft platform for determining plume composition downwind of various volcanoes. I believe that detection of other gases (besides H2O) will be or already have been added to the instrument's capabilities? What other instruments would be useful to run alongside this one? In particular, I'm thinking that it would be quite advantageous to measure SO2 in parallel with the other gases so that emission rates for all species can be derived from remote sensing SO2 measurements.

References: The reference list needs to be expanded to include more relevant publications. I have made a number of suggestions above and additional suggestions are also welcome. Some of the technical documents referenced are pretty much just URLs and might be better referenced in the text itself, depending on what the journal policy is for these.

---

## Referee Comment (RC2) · R. Scholten (Referee) · 6 Oct 2017

The paper is an interesting application of standard absorption spectroscopy techniques. There are a number of specific criteria that the authors address, particular to application for studying volcanic plumes. They have developed an instrument that successfully achieves their requirements for volcanic science, including precision, power consumption, size and weight.

I felt that the manuscript could be of greater value to the community and have greater impact with some relatively minor changes.

[Figure]

First and foremost, the 3D Figure 1 is useful, but would be much more useful if accompanied by a schematic diagram showing the optical configuration. That is, a box and line drawing showing the key elements (lasers, mirrors, detectors, fibres) and the light paths, particularly the multipass arrangement in the cell. Figure 2 would also be much more useful if they could provide line drawings showing waht the cells are - to my untrained eye, I see a skinny black tube and a fat white tube, which provide little guidance if I wanted to build a similar instrument. Scale bars would be helpful for both figures.

Second, there is frequent mention of key metrics such as precision, power consumption, size, weight, but the requirements are not quantified. The manuscript would be much more readable, and also easier to write, if the requirements were specified in the introduction. What precision is needed for their volcanic research? What power consumption is acceptable? How long must it operate on battery power, or what could be achieved if that was possible for 1 hour, 4 hours, 8 hours etc? Why is size important and what size thresholds are there? Once those parameters are defined, it becomes very clear to list alternatives that do not meet those requirements, and show how their instrument compares. The conclusion paragraph could also then be more concise and have greater impact.

In a related sense, the paper is quite long for the content. It would have greater value if more concise; for example removing frequent repetition (references to precision, power consumption, etc). In considering alternatives, they could be tabulated, with a row for each, and columns for the metrics mentioned above. A reader could then very quickly appraise the existing options and see why this new instrument is worth learning about.

I have a number of small queries/suggestions. Line 120: I did not understand this sentence.

Line 150, 228: The manuscript uses mixed units for laser wavelength/frequency (e.g. laser wavelength in microns and spectroscopic features in wavenumbers). It would

be helpful to provide both when using either, for example 1.27um (7875 cmˆ1) and 7823.82 cmˆ-1 (1.27815 um).

Line 182: The data acquisition requirements do not seem very demanding (1 MHz/16-bit measurement), and thus the use of a system crate with dual-core processor and FPGA seems extreme for an instrument which is nominally low power and compact. I am not particularly expert in what is available in such technology but I would first consider something like a BeagleBone single-board computer, raspberry Pi, and other devices which operate at just a few watts. Could they comment on why this approach was taken rather than something more specific to the task?

Line 235: Why did they not use two detectors and a dichroic splitter, so they could detect both wavelengths simultaneously, rather than switching them on and off?

How ere the scanning times determined, i.e. the ramp time of 1600us, and the dead-times of 100us and 300us? What is the time-constant of the detection system? Are these times excessive, or is the 1.6ms ramp time so fast that there is significant loss of fidelity due to scanning too fast over small features that are lost due to the analogue bandwidth?

Line 250: It would be helpful if the authors could provide a calculation of the expected measurement time required to achieve their desired sensitivity, and compare that to the time they used in their experiments. In particular, to better understand the limits; for example, could we reasonably expect to achieve measurements of comparable precision and accuracy at 10/s or 100/s rather than 4/s? If not, what would we need to improve?

Line 257: They refer to simulation of a sloping background by *multiplying* a Voigt profile by a two-order polynomial (which I take to mean a quadratic). But fitting a background would involve addition/subtraction of a polynomial; multiplication simulates a varying gain or optical power. Could the authors please clarify?

[Figure]

Line 330: I did not understand this sentence: what do they mean by "get rid of fringes"? What fringes?

Line 388: The instrument measures ambient $CO_2$ concentration of (390 +/- 20) ppm. How does that compare with standard data or independent measurement?

Line 434 (conclusion): Again, it would be helpful if the authors could compare their achievements against quantitative metrics. Some suggestions on how the instrument might be improved could be helpful (improved in terms of precision, accuracy, power, weight, size etc). I wondered if perhaps an AC measurement technique might be useful; their measurements are essentially DC, and thus sensitive to low-frequency (1/f) noise. Would fast modulation (of frequency or power) and demodulation (i.e. lock-in detection) be helpful?

In summary, the authors have developed a very nice instrument and their work is likely to be useful to others interested in field-deployable spectroscopy, but could be much more useful with some schematic diagrams and a little reorganisation of the manuscript.

---

## Author Comment (AC1) · 10 Nov 2017

**REPLY TO REVIEWER #1**

**SPECIFIC ISSUES**

The manuscript would benefit from a more detailed literature review and inclusion of more relevant references to other, previously published work. In the introduction, it would be helpful to mention all the current methodologies that are typically used to measure HF in volcanic plumes: filter packs, diffusive tubes and chemical traps are missing, along with at least one reference for each. Direct sampling is also a means of measuring HF and $CO_2$, and is mentioned, but no references are given. Clearly, none of these techniques perform the analysis in the field, so they are in some ways inferior to the new instrument described here, but they are the current standard means of performing these measurements and worth describing and referencing.

We agree, this has been addressed through rewriting of the introduction and adding references.

1) In order to give a better description of the instrument, a schematic plot of the apparatus, where the path of light from each of the two lasers can be followed, has been added as Figure1.

2) In order to clarify some aspects of the instrument (as the functionality of the reference signal, the meaning of the "zero-power signal", the characteristics of the raw reference and main spectra, the importance of the fitting procedure), Section 4 has been expanded and re-arranged and Figure 4 (showing the acquired reference and main signals before processing) has been added.
   In particular the role of the reference channel has been clarified with the following sentences:

   "The reference spectrum (bottom side of Fig.4) consists of two kind of signal: the interference fringes due to the etalon and the absorption lines of the two molecules. Known the etalon FSR's, the interference signal allows to obtain a relative frequency reference for the calibration of the frequency scale. The presence of the absorption signals is necessary only to check that the laser emission frequencies don't change over time. This is particularly important for HF, which is usually absent in the atmosphere, since, without reference cell, it is impossible to verify the stability of the laser emission frequency and its overlapping with the selected absorption lines."

   The referee is right when saying that the reference channel is used only for the frequency calibration of the horizontal scale and it is not used directly for the retrieval of $CO_2$ or HF mixing ratio. However the frequency calibration becomes essential to have an absolute value of the concentration without need of calibration and to provide absolute measurements is one of the great advantage of the Direct Absorption technique. Moreover, also the use of the reference cell is important to check the stability of the laser emission frequency. For instances, the laser frequency could change for a failure in the driver of the pump current or for a change in the laser working point. In such cases the emission frequency can be shifted far from the absorption lines and, in particular for HF, which is usually absent in the atmosphere, it is impossible to understand if the laser emission is in the right position only checking the etalon fringes.

3) The mechanism we use to reduce the pressure inside the multipass cell during laboratory test is very trivial: we have only a scroll pump and two needle valves. During the flow of the gas mixture, we adjust the two valves to set the measured pressure at the desired value.
In order to explain better, we have added at the beginning of Section 5 the following sentence:

"For the measurements below room pressure we used a scroll pump and two needle valves, at the entrance and exit of the multipass cell, to set the pressure at the desired value."

4) Finally, it would be useful to include more details on the field experiments that were performed with the instrument at Vulcano Island. Where exactly was the instrument located (perhaps include a map?). More importantly, which fumaroles were being sampled? Vulcano is a well-studied field site. How do the results of the measurements compare with those obtained by others? See e.g. Aiuppa et al 2004, Intercomparison of volcanic gas monitoring methodologies performed on Vulcano Island, Italy and Inguaggiato et al 2012, Total CO2 output from Vulcano island (Aeolian Islands, Italy), but there are many other reports too. While it is not the main thrust of this manuscript to provide new data for volcanology, comparing the obtained results with others would strengthen the case that the instrument is performing as expected.

We agree with the reviewer and we've included the following added paragraph.

Our measurements of $CO_2$/HF with a molar ratio of $570 \pm 30$ were performed downwind of the F0 fumarole on Vulcano, so we expect most of the measured gases to be sourced from here, however we cannot exclude mixing with other fumarolic sources. This allows a comparison with measurements collected with OP-FTIR on F0 fumarole in 2002 (Aiuppa et al., 2004), which revealsed a $CO_2$/HF molar ratio of $175 \pm 20$. This difference may reflect either a change in gas composition from fumarole F0 or a potential contribution from different fumarolic sources in each measurement.

**Minor corrections:**

Please consider that in the following the number of pages, lines and figures refer to the old version of the manuscript.

**Abstract, L4 – Consider changing 'remove all problems to 'mitigate problems associated with chemisorption'**
Done

**P1, L20 – Please give references for examples of direct sampling and sampling via alkali solutions.**
We added several references for all types of sampling systems

**P1, L22 – Consider adding Aiuppa et al 2005, 'Chemical map- ping of a fumarolic field: La Fossa crater, Vulcano island (Aeolian islands, Italy)' and Shinohara (2005), A new technique to estimate volcanic gas composition: plume measurements with a portable multi-sensor system to the references for MultiGAS.**
Done

**P1, L23, consider omitting 'to be performed'**
Done

**P1, L24, 'quantify due to THE slow and differing: : :'**
Done

**P2, L1, consider replacing 'chemical-based' with 'electrochemical'.**
Done

**P2, L2, Please add references for remote sensing via ultraviolet spectroscopy, e.g Galle, B., C. Oppenheimer, A. Geyer, A. J. S. Mcgonigle, M. Edmonds, and L. Horrocks (2002), A miniaturised ultraviolet spectrometer for remote sensing of SO₂ fluxes: a new tool for volcano surveillance, J. Volcanol. Geotherm. Res., 119, 241–254. and Edmonds, M., R. A. Herd, B. Galle, and C. M. Oppenheimer (2003), Automated, high time-resolution measurements of SO₂ flux at Soufriere Hills Volcano, Montserrat, Bull. Volcanol., 65(8), 578–586, doi:10.1007/s00445-003-0286-x. and Galle, B., M. Johansson, C. Rivera, Y. Zhang, M. Kihlman, C. Kern, T. Lehmann, U. Platt, S. Arellano, and S. Hidalgo (2010), Network for Observation of Volcanic and Atmospheric Change (NOVAC)-A global network for volcanic gas monitoring: Network layout and instrument description, J. Geophys. Res., 115, D05304, doi:10.1029/2009JD011823.**
**Also consider adding more references pertaining to IR spectroscopy.**
Done, we added the references: Francis, P., Burton, M. R., Oppenheimer, C., Remote measurements of volcanic gas compositions by solar occultation spectroscopy, Nature, 396, 567-570, 1998, doi:10.1038/25115; Mori, T., Notsu, K., Remote CO, COS, $CO_2$, $SO_2$, HCl detection and temperature estimation of volcanic gas, Geophys. Res. Lett., 24, 2047-2050, 1997

**P2, L3, Please define 'path amounts'. I assume you mean path-integrated concentrations.**
We have replaced "path amount of gas" with "path-integrated concentrations of gas".

**P2, L4, I would argue that the technique described in this paper falls under the category of infrared spectroscopy and therefore is not 'poorly suited' for in-situ measurements.**
We agree with the Referee and we have removed "is poorly suited for in-situ measurement".

**P2, L4, I'm not sure what an 'in-situ spectrometer' is.**
In order to clarify the text, we replaced the sentence: "Tunable diode laser-based in-situ spectrometers may overcome the challenges of chemical sensors, allowing traceable, accurate and precise measurements," with "Tunable diode laser-based spectrometers may overcome the challenges of chemical sensors, allowing traceable, accurate and precise measurements directly on site,"

**P2, L8, Ultraviolet spectroscopy mostly only measures sulfur dioxide. Therefore, it's not clear why sensitivity to multiple gases is an 'essential requirement for volcanic gas sensing'. Perhaps it's essential to in-situ measurements?**
The ratios among different concentrations provides information about the status of the magmatic chamber. But the concentrations of gases in the volcanic plumes is a partial information. In order to quantify the emissions, fluxes must be known instead. It is relatively simple to know the flux of $SO_2$ from a volcan, by using UV cameras. When coupled with this information, the knowledge of the

concentration ratios among $SO_2$ and other gases by in-situ measurements allows to retrieve the fluxes of these gases too.

We added the following sentence: "$SO_2$ is relatively easy to quantify due to a strong UV absorption spectrum and low background concentration, allowing straightforward $SO_2$ flux quantification {Stoiber1983,Galle2003,Mori2006}. So, the knowledge of $SO_2$ flux, when added to the in-situ measurement of the ratios among $SO_2$ and other gases, extends the information about flux to these gases too".

The added references are:

Galle, B., Oppenheimer, C., Geyer, A.,McGonigle, A. J. S., Edmonds, M., Horrocks, L. A., A miniaturised UV spectrometer for remote sensing of $SO_2$ fluxes: A new tool for volcano surveillance, J. Volcanol. Geotherm. Res., 119, 241–254, (2003), DOI={10.1016/S0377-0273(02)00356-6}.

Mori, T., Burton, M. R., The $SO_2$ camera: A simple, fast and cheap method for ground-based imaging of $SO_2$ in volcanic plumes, Geophysical Research Letters, 33, L24804, 2006.

Stoiber, R. E., Malinconico Jr., L. L., Williams, S. N., Use of the correlation spectrometer at volcanoes, in Forecasting Volcanic Events, edited by H. Tazieff, and J. C. Sabroux, pp. 425 – 444, (1983), Elsevier, New York.

**P2, L9, Consider omitting 'whilst avoiding chemisorption processes' here and discuss it later.**
We removed "whilst avoiding chemisorption processes", as it's already discussed later.

**P2, L20, '$CO_2$ is HIGHLY insoluble…'**
Done

**P2, L20, Please clarify what becomes saturated with what at depths larger than 10 km.**
We replaced "$CO_2$ is highly insoluble, and becomes saturated at depths typically > 10 km, while the bulk of HF degassing occurs at very low pressures" with "$CO_2$ is highly insoluble, and begins to exsolve from magma at depths typically > 10 km, while the bulk of HF degassing occurs at very low pressures"

**P2, L28, I believe that quantifying small changes in gas concentration requires high 'precision', not high 'sensitivity', correct?**
We prefer to use "sensitivity". Please see note P9,L1 in the following.

**P2, L29, Consider rewording to '(in order to RESOLVE RAPID CHANGES IN GAS COMPOSITION).**
Done

**P3, L3 – The LICOR LI-840A also measures $CO_2$ and $H_2O$. Consider rewording to 'Several commercial instruments provide simultaneous detection of $CO_2$ and $H_2O$ (e.g. LICOR 7000 and LICOR 840A).**
We agree, done, and the reference has been modified too

**P3, L13 (Environment) is not a valid reference. Please describe FTIR instruments in the text rather than in the reference list.**

We didn't want to describe in general FTIR spectrometers, but only to mention another class of commercial devices. So we modified "In Fourier Transform Infrared (FTIR) spectrometers" in "In commercial Fourier Transform Infrared (FTIR) spectrometers"

**P3, L21, …on the COMMERCIAL market.**
Done

**P3, L22, The design requirements for volcanological applications in general vary quite a bit depending on access to the volcano, instrument deployment platform, volcanic hazards, monitoring vs basic research, and other factors. Perhaps it's best to focus on the design requirements for airborne measurements of volcanic plumes here?**
We agree with the reviewer about the wide set of operating conditions related to volcanic applications. Yet, there are some constraint which apply to all kind of devices: capability of facing harsh environment, low power and weight, low detection limit and high accuracy. Shaking an instrument in a backpack when climbing on Etna or Vulcano, or vibrating it in a helicopter, is just a matter of frequency spectrum of vibrations, but the instrument must be anyway light, with low power consumption and so on. So we prefer to maintain our approach.

**P3, L26, : : : less sensitive THAN CRD or ICOS,: : :**
Done

**P3, L27, Consider rewording to 'source**
Done

**P3, L29, '… multipass cell, optimizes the instrument size and reduces the weight'**
Done

**P3, L32, 'two GAS CONCENTRATIONS and THEIR RELATIVE ABUNDANCE'**
Done

**P3, L34, '…values of the two gases can be provided at a maximum rate of 4 Hz without the need for calibration.'**
Done

**P4, L3, '…laboratory PERFORMANCE of the device'.**
Done

**P4, L3, Consider omitting 'with an Allan-Werle Variance analysis' here and discuss this later.**
We agree, done

**P4, L5, Consider omitting the detection limits here, as these are results and should be reported on later.**
Done, the sentence "Finally, the results obtained during a first test campaign at the crater of Vulcano volcano (Aeolian Islands, Italy) will show an in-field detection limit of 320 ppb for $CO_2$ and of 20 ppb for HF with an integration time of 2 s" has been changed to "Finally, we will show the results obtained during a first test campaign at the crater of Vulcano volcano (Aeolian Islands, Italy)".

**P5, L5, Consider replacing 'exploit' with 'use'**
Done

**P5, L12, '…orientation of the final mirror MAY require OPTIMIZATION.'**
Done

**P5, L15, '… multipass cell ARE sent…'**
We have preferred to use "beam" instead than "beams"

**P5, L24, 'All electronics ARE placed…'**
We have changed the sentence in "The electronic part of the instrument is placed aside the cell…"

**P5, L25, Please explain the acronym FPGA**
Done

**P5, L30, 'In order to protect THE optics and electronics from volcanic gases…'**
Done

**P5, L34, 'So by neglecting this ADDITIONAL path, we overestimate the ambient concentration of CO2 by 0.6% or about 2 ppm. The relative effect is smaller when measuring $CO_2$ concentrations above ambient.'**
**P5, L34, Do you overestimate the path by 0.6%? If you know about this issue, why not simply correct for it?**
As said previously, in order to protect the optics against the volcanic gases, they are enclosed within a sealed plastic cover, so that the air inside this cover is not the same air measured between the mirrors of the multipass cell. Consequently we don't know exactly how much we overestimate the CO2 concentration, we can only affirm that our measurement can be overestimated of about 2 ppm when in the multipass cell there is ambient CO2 and that the effect becomes lower when the air in the multipass cell contains a higher CO2 concentration. To explain that, we have changed the sentence as suggested by the referee: "So by neglecting this additional path, we overestimate the ambient concentration of CO2 by 0.6% or about 2 ppm. The relative effect is smaller when measuring CO2 concentrations above ambient".
Moreover it is impossible to measure definitevely the CO2 concentration inside the cover (for instance by flowing nitrogen into the multipass cell), because this value should be different each time we open and close the cover to adjust the optics.

**P6, L6, 'the 1 liter volume'**
Done

**P6, L12, '4 cell LiPo batteries'**
Done

**P6, L13, '… a weight of about 8 kg (pump and batteries included), WHICH makes it particularly suitable as a portable instrument for in-situ operation in a hostile environment such as in a volcanic area.'**
Done

**P6, L15, 'unattended and be remotely controlled via WiFi from outside the area of toxic gas emission.'**
Done

**P6, L16, How far does the WiFi actually reach?**

We do not consider the distance at which we can communicate via WIFI with a instrument a fundamental feature of our sensor, so we prefer not to give a precise number. In fact, this distance changes with the characteristics of the area in which the measurements are made and can be easily extended by choosing a different commercial component.
Usually we do not exceed 10-20 meters, a distance that gives us the opportunity to work in a more comfortable environment.

**P6, L20, '… across a characteristic absorption LINE of the target molecules.'**
We have changed the sentence in "…across the selected absorption line of the target molecules"

**P6, L21, '… HF absorption LINE…'**
Done

**P7, L2, replace 'alternatively' with 'alternatingly'**
Done

**P7, L5-6, Replace 'a region' with 'an interval'**
Done

**P7, L8, '… around the CO2 line and 1.5 cm-1 around the HF line, respectively.'**
Done

**P7, L10, Please clarify what is meant by the 'zero-power signal'. I assume you mean the intensity measured on either side of the absorption line?**
No, we mean the signal acquired by the detector when the laser power is zero. This signal is acquired during the first interval of the ramp when the laser is switched off. In order to clarify this point we have added a sentence in the description of the ramp:

"As shown in Fig. 4, the modulation signal of each laser consists of 3 parts: (i) an initial interval, with a duration of 100 µs, during which the laser is turned off to get the background signal of the detector when no laser power is incident on it (in the following we refer to this signal as "zero-power signal"); (ii)… "

To describe the importance of the zero-power signal we have expanded a previous sentence in the following:

"For each acquisition the zero-power signal can be subtracted, so that the absorbance can be derived independently from the absolute laser power. Consequently we don't need to know exactly the splitting ratio of the beam splitter or any kind of variability in the laser power. Moreover the changes of reflectivity of the mirrors in the multi-pass cell, related to the interaction with the external ambient, do not influence the measured absorbance value except for affecting the signal to noise ratio."

**P7, L10, 'allows derivation of the absorbance independent of the absolute laser power. Consequently, the splitting ratio of the beam splitter and the reflectivity of the mirrors in the multi-pass cell do not influence the measured absorbance value except for affecting the signal to noise ratio.'**

For the changed sentence, see the previous note

Differently from the changes in the reflectivity of the mirrors, the splitting ratio of the beam splitter don't affect the signal to noise ratio.

**P7, L114, What is a '4000-points main signal'? Are you sampling the spectrum with 4,000 points per wavelength scan?**

Yes. To better clarify, we have described the acquired scan at the beginning of the Section and in particular we have added the sentence:

"The two signals measured by the main detector (D1) and by the reference detector (D2) are acquired synchronously on 2 acquisition channels of the CompactRIO at 1 MSample/s with a resolution of 16 bits. The two main and reference signals, sampled each with 4000 points per scan, are averaged 25 times and saved for a post-processing. Typical in-field main and reference signals are shown in Fig. 4."

**P7, L16, I understand that the measurement of $CO_2$ and HF is nearly coincident for practical purposes of the measurement, but I would omit 'simultaneous' here because you just explained that the two gases are measured in alternating manner.**

Done

**P7, L16, Omit 'of' before 0.25 s.**

Done

**P7, L18, Again, please clarify what 'zero-power signal' means**

Done (see previous note P7, L10).

**P7, L18, '…ARE fit with… multiplied by a second order polynomial.'.**

Done

**P7, L18, Are you sure you need to multiply by a second order polynomial? I would have expected the addition of a polynomial to described the ramping intensity. As mentioned before, it would be nice to actually plot a raw spectrum that includes the interference fringes (in the reference channel) and the ramping up of the laser intensity.**

In order to clarify the point, the absorption signal acquired by the main detector is shown in Fig. 4. The frequency scan is obtained by modulating the laser current, so also the output power of the laser results modulated according to the ramp and, without absorption, the signal should be described by a second-order polynomial. The absorption spectrum is the exponential of a Voigt profile multiplied for the polynomial describing the laser power. The polynomial is not a background signal but only the shape of the laser power.

To remove misunderstanding we have changed the sentence:

"…multiplied by a second-order polynomial, which simulates the sloping background due to the ramping of the driving current."

with :

"…multiplied by a second-order polynomial, which simulates the sloping laser power due to the ramping of the driving current."

**P7, L22, which 'molecular parameters' from the HITRAN database are relevant? Can you please be more specific?**
The HTRAN molecular parameters used in this calculation are specified:

"The values of the HWHM are calculated as a function of temperature and pressure, measured for each acquisition, and of the molecular parameters (airbroadening coefficient and linewidth temperature coefficient) according to the HITRAN database"

We would like to stress that all the details connected with the data processing (containing formula, parameters, error calculation etc.) were already described in a previous paper (Ref. Viciani et al. Appl. Phys. B, 90, 581-592, 2008), consequently we prefer not to give so many details in this paper. To emphasize this point we have added the following sentence:
"Here we present a quick description of the data processing, since a more detailed analysis has already been reported (Viciani et al. 2008)."

**P7, L25, 'multiplied BY a SECOND order…"**
Done

**P7, L20-27, The HITRAN database contains line strength information, not absorption cross-sections. The line strength is defined as the integral of absorbance of a given line. Since this TDL instrument resolves the individual absorption lines of $CO_2$ and HF, respectively, couldn't the line strength simply be measured as the integrated absorption over the measured wavelength interval? The advantage of deriving the line strength directly is that it should be independent of temperature and pressure, correct? Could you please explain why the four-Lorentz Puerta-Martin approximation is used rather than simply determining the line strength and calculating the column density and gas concentration from that?**

According to the Beer–Lambert law, the intensity signal $I$ measured after the multipass cell at a given optical frequency $v$ is a function of the laser intensity $I_0$, of the gas cross-section $\sigma$, of the optical path length $L$ associated with the multipass cell, and of the concentration $N$:
$I = I_0\, exp[-\, N\, L\; \sigma(v)\,]$

If we integrate the measured absorbance, we obtain:
$\int\; [-\, ln(I/I_0)]\; dv\; =\, N\, L\; S$

where $S$ is the line-strength. So, we can infer $N$ from the integrated absorbance only if we know also $S$. In our fitting procedure (described in detail in Ref. Viciani et al. 2008), we fit the measured signal $I$ with the function:
$P(v)*Exp[-\, V(v,\, fitting\; parameters)]$

where $P(v)$ is a second-order polynomial and $V(v)$ is a Voigt profile approximated with the four-Lorentz Puerta-Martin function. With the parameters obtained by the fit, we calculate the integrated Voigt profile and:

$N = \int V(v) \, dv \, / \, (L \, S)$

This fitting procedure has 2 advantages with respect to the direct integration of the absorbance:
(i) We don't need to know exactly $I_0$ (because it is one of the fitting parameter).
(ii) The result of the integration of an experimental signal is very noisy because affected not only by the fluctuation of the signal but also by the fluctuation of the background outside the absorption. Differently the he result of the integration of a Voigt function is less noisy and not at all affected by the background.

**P8, L5, '…laboratory performance…'**
Done

**P8, L5-10, How do you know that chemisorption prohibits a laboratory test using a pre-mixed HF calibration gas? Did you attempt the experiment? Can you provide a reference from the literature as to why such an experiment would surely fail? I would think that dust and water vapor in the sampling apparatus could be avoided in a laboratory setting.**
In order to do such a test with HF, we need a gas line including: a tank with a calibrated HF mixture, a pressure regulator, a Teflon pipe from the regulator to the multipass cell and the  closed teflon cell. To have a reliable measurement, all the surfaces of these components should be completely free from dust and water vapor, otherwise they will be involved in chemisorption. We think this is a very difficult issue to obtain also in a laboratory! We think that it is more reliable to evaluate the performance of the HF channel in an open-cell configuration.

**P8, L11, '… performance of the HF channel…'**
Done

**P8, L12 and throughout the manuscript: Consider running a global 'search and replace' on the entire manuscript and replace all instances of 'has been' and 'have been' with 'was' and 'were' unless referring to studies that were conducted previous to the work presented here.**
We agree, all "has been" and most of "have been" were replaced

**P8, L14, How was the under-pressure of 700 mBar achieved?**
We have added in the text the following explanation sentence:

"For the measurements below room pressure we used a scroll pump and two needle valves, at the entrance and exit of the multipass cell, to set the pressure at the desired value."

**P8, L17, I don't understand why two optical fringes need to be included in the fit procedure when fitting the measurement channel. The reference channel passes the etalon, so it makes sense that fringes would appear there, but why are they included in the fit for the measurement channel? Also, they are not obvious in figure 3. Is that because they are very low-amplitude compared to the absorption line?**
In all optical instruments there are  unwanted reflections among optical surfaces which generate spurious interference fringes, usually known as "optical fringes". Often in Direct Absorption Technique with multipass cell the sensitivity of the instrument is limited by the optical fringes (see as an example

A. Fried et al., Applied Optics 29, 900-902, 1990; or V. V. Liger Spectrochimica Acta Part A: Molecular and Biomolecular Spectroscopy 55, 2021-2026, 1999).

A standard method to reduce the effect of the fringes is to include in the fitting function one or more sinusoidal curves to get rid of these fringes (see Ref Viciani et al. 2008). In Figure 5 there is no evidence of the fringes because their amplitudes are of the order of $10^{-3}$, but without the two optical fringes in the fitting function the residual should show a sinusoidal behavior with amplitude $10^{-3}$. Obviously the effect of the optical fringes becomes relevant for low concentration values.

To explain better we have added in Section 4, in the description of the fitting function, the following sentence:

"When necessary, also one or more sinusoidal curves are included in the fitting function to take into account the presence of optical fringes."

**P8, L28, Please replace 'lower pressure' with the actual pressure that was used (700 mBar?)**
Done

**P8, L30, Please explain why a narrower line shape improves the identification of the background intensity, and why optical fringes are present in the measurement spectrum.**
For the presence of optical fringes see note P8,l17.
As already said, the function used to fit the signal in Figure 3 (top) is given by:
$P(v)* Exp[- V(v, fitting\ parameters)] * OF(v)$

where $P(v)$ is a second-order polynomial which describes the background, $V(v)$ is the approximation of a Voigt profile and $OF$ is a sinusoidal curve which describes the Optical Fringes.

For a broad lineshape, as wide as the laser scan (as that one of Figure 5), the exponential part is prevalent with respect to the other two terms, and it results very difficult to determine the fitting parameters for $P(v)$ and $OF(v)$. Consequently, repeated fitting procedure are affected by a variability in the determination of $P(v)$ and $OF(v)$. This variability is reduced when the lineshape becomes narrower because $P(v)$ and $OF(v)$ can be better identified by the fit.

In order to better clarify this point, we have added the following sentence:

"The lower precision at ambient pressure is due to the fact that the absorption signal is as broad as the laser scan and it is not easy for the fitting procedure to clearly identify the parameters of both the second-order polynomial which describe the background and the optical fringes. When the pressure is reduced and the absorption signals are narrower, the fitting protocol becomes more precise."

**Figure 3 caption, '…in the bottom plot…'**
Done

**Figure 3 caption, and throughout the manuscript: When taking the average of multiple acquisitions to improve the signal to noise ratio, the term 'integration time' is typically used to describe the total time for all the individual acquisitions. The 'acquisition time' usually describes the time for just one acquisition. Consider using this verbiage throughout the manuscript.**
Done in all the text of the paper.

**P9, L1, 500 ppb appears to be the precision of the instrument, not the sensitivity of the instrument.**

We apologize for the confusion between precision, sensitivity and detection limit.
We have corrected the text according to the following formalism:
The PRECISION is the stability of the instrument.
The DETECTION LIMIT is the minimum concentration which the instrument can measure and it is given by the acquired signal for which the signal-to-noise ratio S/N = 1.
The SENSITIVITY is the minimum variation in the concentration detectable by the instrument.

The sensitivity can not be better than the precision, but in principle it could be worst.
In our instrument the sensitivity is determined by the precision.
So the "500 ppb" is the precision and also the sensitivity.
We have maintain the term "sensitivity", but to clarify this point we have added the following sentence:

"In the present case, the sensitivity of the instrument, defined as the minimum variation in the mixing ratio detectable by the instrument, is entirely determined by the precision. In order to evaluate the ultimate sensitivity of the $CO_2$ channel, an Allan-Werle Variance analysis of the obtained mixing ratios was carried out."

**P9, L2, Detection limit and precision are not quite the same thing. In this case, since you will always have at least 400 ppm $CO_2$ in the ambient air, the detection limit is not really relevant I guess. Instead, the precision is what will determine if a small volcanic plume can be detected or not.**
We have replaced detection limit with sensitivity (see also note P9, L1).

**P9, L6, please explain what is meant by 'get rid of fringes'.**
See notes P8,L17 and P8,L30.
In order to explain better we have changed the sentence:

"Again, we believe that this reduction of the best integration time at low pressures is due to the better capability of the fitting protocol to clearly identify the background signals and the optical fringes."

**P9, L12, Please include more details on where the measurements were made, perhaps even a map?**
**See reply to point 4 above**

**P10, L6, Consider replacing 'emission peak' with 'gas cloud'**
Done

**P10, L7, '…the only way to completely solve this problem…:'**
Done

**Figure 5 caption, '…is due to chemisorption on instrument components in the closed cell configuration.'**
Done

**Figure 6 caption, 'bottom plots'**
Done

**Figure 6 caption and throughout the manuscript: Please be a little bit more precise when using the term 'concentration'. Concentration is a measure of the number of trace gas molecules in the sample cell or in a known volume of air. A typical unit is molecules / cm$^3$. A mixing ratio is the ratio of trace gas molecules to total air molecules in a given volume. Typical units are ppm or ppb. The TDL instrument measures absorbance, which can be converted into column density (units of molecules / cm$^2$) using the Beer Lambert Bouguer Law. This can further be converted into an average concentration in the cell. However, converting to a mixing ratio requires knowledge of temperature and pressure, as described in your equation 1. Since you are measuring temperature and pressure and correcting for these effects, it's fine to report mixing ratios, but please don't call them 'concentrations.'**

We apologize for that. We have replaced "concentration" with the "mixing ratio" in the text of the manuscript, in the captions of Fig. 5, 6,7 and 8 and in the axes of Fig.7

**P12, L6-8, Please clarify how you obtain the mean value of the noise from the residual. It appears that the residual contains some systematic structures as well as some statistical noise. How do you separate the two? Do you have any ideas where the systematic structures come from?**

The structures in the residual are due to the presence of optical fringes (see note P8,L17), which have not been removed completely by the fitting protocol. The important point is that this structure is different for each spectrum, due to the random fluctuation of the phase of the optical fringes. So it can not be considered as a systematic structures. For different spectra the structure of the residual is different, but the standard deviation is very similar. So a typical noise value associated to the measurement can be inferred by the mean of the noise of different spectra.

To avoid misunderstanding we have removed the sentence: "most of which show a residual similar to that one displayed in Fig.6 ".

**P12, L10, Again, do you mean precision rather than detection limit?**

We have changed "detection limit" with "sensitivity". Please see note on Figure8 in the following and also note P9,L1.

**P12, L13, '... a factor of 1.3'**

Done

**Figure 7 caption, Again, this figure plots mixing ratios, not concentrations.**

Done

**Figure 7 caption, '... a 3-minute interval.'**

Done

**P13, L6, Perhaps rephrase this sentence to 'If the instrument is operated in lower acid gas and aerosol concentrations it could likely run for long periods of time without significant degradation of the signal to noise.'**

We have replaced:

"On the contrary, when the instrument is far from the fumaroles, or on board an aircraft, it can work indefinitely without significant reduction in performance."

with:

"On the contrary, if the instrument is operated in lower acid gas and aerosol concentrations, as for instance on board an aircraft, it could run for long periods of time without significant degradation of the signal to noise ratio."

**P13, L14, Do you mean 3 per mil or 3 percent? Since percent are used throughout the manuscript, consider replacing this with '0.3%' if that is the appropriate value.**
We mean 0.3% and we have replaced it.

**P13, L15, 'emission peak could be used to correct'**
Done

**Figure 8, again plots mixing ratios rather than concentrations**
Done

**Figure 8, If I understand correctly, then the estimated detection limit of the instrument for $CO_2$ was determined by extrapolating the measurements shown in Figure 8 a down to a S/N of 1. If this is true, this estimate would appear to have quite a large uncertainty given that no measurements were performed with S/N less than about 1800. If you want to peruse this methodology, please give the uncertainty of the detection limit obtained in this manner. In my opinion this is of somewhat limited use, however, as no measurements will ever be performed at less than about 400 ppm $CO_2$ which corresponds to a S/N of more than 1,000. For the $CO_2$ channel, the precision of the instrument is much more important than its detection limit.**
Again we apologize for the confusion.
In Figure 8 we show the linear relationship between S/N and MR (mixing ratio), in order to find the slope through a linear fitting procedure.
According to the formalism in note P9,L1:
- the detection limit is the MR for which S/N = 1.
- the sensitivity is the variation in the mixing ratio corresponding to a variation of the signal equal to the noise. If S/N is a function F of the MR, the sensitivity (for a particular MR_0) is the inverse of the derivative of F with respect to MR, calculated in MR_0.
In the present case, where there is a linear relationship between S/N and MR, the detection limit and the sensitivity have the same value equal to the inverse of the slope.
Of course, we agree with the Referee that  the detection limit for CO2 is not an important parameter for our instrument. So we have replaced "detection limit" with "sensitivity"

**P14, The manuscript lacks a 'Conclusions' section. The first paragraph on page 14**
**seems to belong to the conclusions so this might be an appropriate place for this heading.**
Sorry for that, during the submission procedure we deleted accidentally the title of the "Conclusion". section. Now we have reinserted it.

**P14, L8, '… by detecting water absorption…'**
Done

**P14, The manuscript only includes a single sentence about future work. Given that this document only describes the very first tests of a new instrument, I would expect much more future work planned for the system. For example, I'm sure the system is to be (or actually already was) run on an aircraft platform for determining plume composition downwind of various volcanoes. I**

**believe that detection of other gases (besides H₂O) will be or already have been added to the instrument's capabilities? What other instruments would be useful to run alongside this one? In particular, I'm thinking that it would be quite advantageous to measure SO₂ in parallel with the other gases so that emission rates for all species can be derived from remote sensing SO₂ measurements.**

According to the suggestion, we have included the following sentence at the end of the Conclusion:

"Moreover we are designing a portable experimental platform for in-situ simultaneous measurements of 5 volcanic gases (HCl, $CO_2$, HF, $H_2O$ and $SO_2$). The platform will include two mid-IR spectrometers, one of which will be the instrument described in this paper, and an UV spectrometer. The platform will be employed for measurement campaigns in-field and on board aircrafts.

In order allow deployment on board of drones, we will reduce size and weight by replacing the cRIO crate with a smaller electronics, namely Red Pitaya by StemLab, we will use only fiber-coupled lasers in the near infrared, so eliminating most mirrors and their holders, and we will use aluminum and carbon fibers only for those parts which strictly require mechanical hardness. All other parts will be realized by 3D plastic printing. As drones can fly much closer to plumes than manned aircraft, the concentrations to be measured are expected to be at least one order of magnitude higher than in the present measurements, and so the multipass cells will be shorter and lighter too."

**References: The reference list needs to be expanded to include more relevant publications. I have made a number of suggestions above and additional suggestions are also welcome. Some of the technical documents referenced are pretty much just URLs and might be better referenced in the text itself, depending on what the journal policy is for these.**

The list of references has been expanded significantly, following the reviewer's suggestions. As for the citations of commercial devices, which refer to URLs, we moved the citations into the text.

**REPLY TO REVIEWER #2**

**First and foremost, the 3D Figure 1 is useful, but would be much more useful if accompanied by a schematic diagram showing the optical configuration. That is, a box and line drawing showing the key elements (lasers, mirrors, detectors, fibres) and the light paths, particularly the multipass arrangement in the cell.**

See point 3) of the reply to reviewer #1

**Figure 2 would also be much more useful if they could provide line drawings showing waht the cells are - to my untrained eye, I see a skinny black tube and a fat white tube, which provide little guidance if I wanted to build a similar instrument. Scale bars would be helpful for both figures.**

Now Figure 2 is Figure 3. We have modified it, removing the closed cap version, and including two photos showing the spots of the laser onto the mirrors. Photo b) shows the whole cell in the alignment setup, and photo c) is a detail of the entrance/exit mirror. The caption has been changed from "Home made multipass cell in the open-cell configuration (a) and in the closed-cell configuration (b)." to "Home made multipass cell in the open-cell configuration (a). Photo b) shows the whole cell with the spots of the alignment laser onto the mirrors, photo c) is a detail of the entrance/exit mirror." We also added a reference to Herriott cells. The scale is given in the figure.

**Second, there is frequent mention of key metrics such as precision, power consumption, size, weight, but the requirements are not quantified. The manuscript would be much more readable, and also easier to write, if the requirements were specified in the introduction. What precision is needed for their volcanic research? What power consumption is acceptable? How long must it operate on battery power, or what could be achieved if that was possible for 1 hour, 4 hours, 8 hours etc? Why is size important and what size thresholds are there? Once those parameters are defined, it becomes very clear to list alternatives that do not meet those requirements, and show how their instrument compares. The conclusion paragraph could also then be more concise and have greater impact.**

We agree with the reviewer and we've added the text below to the introduction, to better explain the requirements for a field-based volcanic gas sensor.

This platform would allow multiple gases to be measured simultaneously at high frequency (at least 2-3 Hz) to permit airborne measurements, would have high precision and accuracy to allow dilute plumes to be measured, and would be compact, low-power and robust to allow easy field deployment with a variety of transport solutions, including backpacks, airplanes and drones.

We have also added this text to the section on state of the art in volcanic gas measurement

In addition to these requirements, the detection of acid gases, such as HF, require further precautions due to the rapid chemisorption of acid molecules on surfaces of the instrument, precluding pumps and filters. Typically, the limits on what makes an instrument field-portable on a volcano are the carrying capacity of a group of two or three people. Thus, up to 10 kg for an instrument and a few kg for batteries is typically ideal.

**In a related sense, the paper is quite long for the content. It would have greater value if more concise; for example removing frequent repetition (references to precision, power consumption, etc). In considering alternatives, they could be tabulated, with a row for each, and columns for the metrics mentioned above. A reader could then very quickly appraise the existing options and see why this new instrument is worth learning about.**

We acknowledge this remark. Wherever possible, we removed redundancies. We don't report here the list of changes, as it would be too long. As for tables, this option could have been exploited when dealing with commercial devices. Nevertheless, we realized that the table should have featured too many columns, so being hard to read, and we left the descriptions in the text.

**Line 120: I did not understand this sentence.**

We replaced the sentence "The latter was a source of concern as we will have the option of using the optical cells in open air" with "When measuring in the plume of a volcan, cleanliness of the mirrors is a serious concern"

**Line 150, 228: The manuscript uses mixed units for laser wavelength/frequency (e.g. laser wavelength in microns and spectroscopic features in wavenumbers). It would be helpful to provide both when using either, for example 1.27 μm (7875 cm$^{-1}$) and 7823.82 cm$^{-1}$ (1.27815 μm).**

We have added everywhere the missing information

**Line 182: The data acquisition requirements do not seem very demanding (1 MHz/16- bit measurement), and thus the use of a system crate with dual-core processor and FPGA seems**

extreme for an instrument which is nominally low power and compact. I am not particularly expert in what is available in such technology but I would first consider something like a BeagleBone single-board computer, raspberry Pi, and other devices which operate at just a few watts. Could they comment on why this approach was taken rather than something more specific to the task?

The crate cRIO by National Instrument is a good compromise for the task to be carried out, and features some important characteristics, namely it comes with all the drivers for LabVIEW environment, and it is possible to realize custom plug-ins, just like we did to supply and stabilize the temperatures of our lasers.

It is possible to add to a Raspberry a 4 input Analog to Digital Converter, with the same resolution (16 bit, see https://learn.adafruit.com/adafruit-4-channel-adc-breakouts/overview), but with at most 3300 samples per second. BeagleBone has a built in set of ADCs, but with 12 bit resolution. We are exploring the Red Pitaya board, whose ADCs feature 14 bits at 125 MS/second. It's a very interesting device, light and with low power consumption, but programming it is quite a complicate task, especially when compared to cRIO.

**Line 235: Why did they not use two detectors and a dichroic splitter, so they could detect both wavelengths simultaneously, rather than switching them on and off? How were the scanning times determined, i.e. the ramp time of 1600 μs, and the deadtimes of 100 μs and 300 μs? What is the time-constant of the detection system? Are these times excessive, or is the 1.6ms ramp time so fast that there is significant loss of fidelity due to scanning too fast over small features that are lost due to the analogue bandwidth?**

As a matter of fact we should have used four detectors and two dicroic splitters, as we should have treated in the same way both the measurement arm and the reference arm. The suggestion is reasonable indeed, but it would complicate the instrument significantly. As for timing, there are several points to be taken into account. First of all, we need some hundreds ramps per second (250), in order to average out white noise. Then, we need a sufficient time (300 μs) to stabilize the laser temperature after switching on. We also need some tens of point to retrieve a good zero value (100 μs), and as many points as possible for fitting the waveform and the etalon fringes (1600 μs). The set values are simply a good compromise among all these constraints. The bandwidths of the laser power supply and of the detectors are not an issue.

**Line 250: It would be helpful if the authors could provide a calculation of the expected measurement time required to achieve their desired sensitivity, and compare that to the time they used in their experiments. In particular, to better understand the limits; for example, could we reasonably expect to achieve measurements of comparable precision and accuracy at 10/s or 100/s rather than 4/s? If not, what would we need to improve?**

We would analyze this point from a different perspective. We want to save all the waveforms produced by the instrument for off-line analysis, though there is an on-board program providing a quick-look of the data. Our processing system takes about 150 ms to treat one waveform, which means that no more than 6 waveforms can be saved per second. Each waveform can be either a single shot, or the average of a variable number of ramps. With the numbers we set, we have a good compromise between time "wasted" and quality of data. For further information about quality of data please refer to the chapter about "Laboratory performances of the instrument for the $CO_2$ channel".

**Line 257: They refer to simulation of a sloping background by *multiplying* a Voigt profile by a two-order polynomial (which I take to mean a quadratic). But fitting a background would**

**involve addition/subtraction of a polynomial; multiplication simulates a varying gain or optical power. Could the authors please clarify?**
Please see the reply to point P7,L18 of Reviewer #1

**Line 330: I did not understand this sentence: what do they mean by "get rid of fringes"? What fringes?**
When interference fringes are due to thin optical elements, like windows, their period is comparable with the whole laser frequency scan. So, if the absorption line occupies a wide portion of the laser scan, it's difficult for the software to correctly identify the contribution of these fringes. The lower the pressure, the narrower the absorption line, the wider the portion of the scan free from absorptions, which can be used for a proper fit of the fringe. An alternative could be that of increasing the laser scan, provided that the overall tunability of the laser allows it. Please see the reply to point P9,L6 of Reviewer #1.

**Line 388: The instrument measures ambient $CO_2$ concentration of (390 ± 20) ppm. How does that compare with standard data or independent measurement?**
We inserted in the text a reference to the trend global data for $CO_2$ concentration issued by NOAA. We added the sentence: "These data fit well with the global trend for CO2 concentration, see for instance the NOAA website (URL = https://www.esrl.noaa.gov/gmd/ccgg/trends/global.html)".

**Line 434 (conclusion): Again, it would be helpful if the authors could compare their achievements against quantitative metrics. Some suggestions on how the instrument might be improved could be helpful (improved in terms of precision, accuracy, power, weight, size etc). I wondered if perhaps an AC measurement technique might be useful; their measurements are essentially DC, and thus sensitive to low-frequency (1/f) noise. Would fast modulation (of frequency or power) and demodulation (i.e. lock-in detection) be helpful?**
We added in the text a further explanation of the reason for our choice of a DC technique: In principle, detection limits and accuracy could be increased of one order of magnitude when adopting a detection technique based of laser frequency modulation (two references have been inserted at this point). Yet, this kind of detection raises severe concerns about calibration, in particular in an environment where temperature, pressure and mixture composition can vary we always aim to versatility. This is because in these techniques the calibration relies on the stability of the lineshape which, on the contrary, is affected by the physical conditions of the measurement. So, also next development of this instrument will adopt direct absorption.
As for other features, we added this paragraph to Conclusions: "In order allow deployment on board of drones, we will reduce size and weight by replacing the cRIO crate with a smaller electronics, namely Red Pitaya by StemLab, we will use only fiber-coupled lasers in the near infrared, so eliminating most mirrors and their holders, and we will use aluminum and carbon fibers only for those parts which strictly require mechanical hardness. All other parts will be realized by 3D plastic printing. As drones can fly much closer to plumes than manned aircraft, the concentrations to be measured are expected to be at least one order of magnitude higher than in the present measurements, and so the multipass cells will be shorter and lighter too".

---

## Author Response (AR2)

**Reply to comments by the Associated Editor**

**First, reviewer #1 was correct that you should not have used the word "alternatively" in this context, but their suggestion of "alternatingly" is not actually a word in the English dictionary! The correct word is "alternately".**
Done

**In your response to the reviewer's comment on various commercial instruments and their URL's, the overrun of the URL on page 3, line 20 means information needed to actually use the URL is missing.**
**In any case it is better not to give the full URL for products as its lifetime is limited due to manufacturers frequent reorganisation of their websites. It is better to limit the URL to just www.company.com since that part of the URL is much more long lived, and it is usually easy to find the product from the top level. Also the text "URL=" is superfluous.**
We found the instruction:
%% URLs and DOIs can be entered in your BibTeX file as:
%%
%% URL = {http://www.xyz.org/~jones/idx_g.htm}
%% DOI = {10.5194/xyz}
at the end of the template. It refers to the BibTeX file, but we thought it should be valid for the text too. Moreover, we asked the Editorial Support about URLs, and we got this reply: "Long URLs can be added to the text without problems. They break automatically to the next line when they are too long and it is better for readers to see the complete link instead of searching on the websites".
But this is just to explain why we used this format. We modified the URLs according to your own instructions. Only NOAA citation has been left with the complete URL, to help the readers.

**On page 4, line 18, the notion that the fibre coupler is a "standard one" is a bit unclear. This statement could be omitted and the rest of the sentence merged with the preceding sentence.**
Whe have merged the two sentences in this way: "(Thorlabs 10202A-50-APC) which, in principle, is not dichroic for the two laser wavelengths".

**Please have a look at page 3, lines 10-15. This reads awkwardly - the passage seems to restart at the second use of the word "several", though in fact this is not the case.**
This part has been rephrased in this way: "Several analyzers fulfill the requirements for $CO_2$ and $H_2O$ measurements, from both academia (Gianfrani et al., 2000; Rocco et al., 2004) and industry (e.g. LICOR 7000 and LICOR 840A, www.licor.com)"

**It is bad form to start a sentence with a symbol - see page 1, line 24 and elsewhere. Page 3, line 17 is how it should be done.**
This and another occurrence have been corrected

**On page 6, line 3, you send a beam \*through\* a cell rather than across.**
Done

**On page 6, line 23 and following, it appears as if some paragraphs may have been meant to be joined. Please be consistent throughout with the line indenting of paragraphs, and whether you have a blank line between paragraphs.**
Blank lines have been removed here, and in Page 3, line 16.

**On page 7 line 10, you should omit the word two - it creates confusion as the two are the main and reference (I think). If in fact you mean there are two main signals then further explanation is needed.**
The word "two" has been removed

**Page 8, line 4; the end of the sentence is tautological**
We removed " for the purposes of our applications"

**Page 8, line 13; "Knowing", not "Known"**
Done

**Page 10, line 11; the commonly accepted symbol for standard deviation is sigma, not 2*sigma.**
In order to be conservative, we used twice the standard deviation as a reference. The sentence has been changed to :"Assuming as precision two times the standard deviation sigma of the …."

**Page 12, line 8; The sentence starting "Moreover …" is unclear in view of what is written later. Is this the value of the correlation after a 16 s time shift has been applied to one of the data series?**
We changed the sentences:
"Correlation analysis of the two concentrations clearly shows that a maximum correlation between the two gases is reached for a delay of about 16 s. Moreover, the correlation is poor with a maximum correlation coefficient of only 30%"
to:
"The analysis of concentrations clearly shows that a maximum correlation between the two gases is reached for a delay of about 16 s. Moreover, even if this delay is considered, the correlation is poor, with a coefficient of only 30%"

**Page 13, line 4; Saying "Three optical fringes … are included" is ambiguous. Do you actually mean three fringes, or three fringe patterns each with different fringe spacing? I would actually expect the latter, and in the former case forcing that number to be an integer is very artificial.**
The right meaning is the latter indeed. We changed the sentence into: "Three optical fringe patterns for $CO_2$ and two ones for HF are also included in the fitting curve".

**page 14, line 2; it's a 3 minute interval, even though it lasts 3 minutes!**
As a matter of fact, we realized that the inset refers to a time interval of 90 seconds. So we corrected text and caption accordingly.

**Page 15, line 6; Why is this in the plural?**
The sentence has been rephrased with singular

**Page 16, line 3; "The air can be taken a few meters apart …" is actually meaningless and needs to be rephrased**
We modified the sentence into: "The air can be taken far from the high concentrations…"

**Page 16, line 17; replace "one" with "absorption resonance".**
We removed "one", but we would prefer "absorption line", rather than "absorption resonance"

[revised manuscript text omitted]